# A-Train estimates of the sensitivity of the cloud to rain water ratio to cloud size, relative humidity, and aerosols

Kevin M. Smalley[1] and Anita D. Rapp[1]

[1]Department of Atmospheric Sciences, Texas A&M University, College Station, Texas

**Correspondence:** Kevin M. Smalley (ksmalley@jpl.nasa.gov)

**Abstract.** Precipitation efficiency has been found to play an important role in constraining the sensitivity of the climate through its role in controlling cloud cover, yet understanding of its controls are not fully understood. Here we use CloudSat observations to identify individual contiguous shallow cumulus cloud objects and compute the ratio of cloud water path to rain water (WRR) path as a proxy for warm rain efficiency. Cloud objects are then conditionally sampled by cloud-top height, relative humidity, and aerosol optical depth (AOD) to analyze changes in WRR as a function of cloud size (extent). For a fixed cloud-top height, WRR increases with extent and environmental humidity following a double power-law distribution, as a function of extent. Similarly, WRR increases holding average relative humidity at or below 850-mb constant. There is little relationship between WRR and AOD when conditioned by cloud-top height, suggesting that once rain drop formation begins, aerosols may not be as important for WRR as cloud size and depth. Consistent with prior studies, results show an increase in WRR with sea surface temperature. However, for a given depth and SST, WRR is also dependent on cloud size and becomes larger as cloud size increases. Given that larger objects become more frequent with increasing SST, these results imply that increasing precipitation efficiencies with SST are due not only to deeper clouds with greater cloud water contents, but also the propensity for larger clouds which may have more protected updrafts.

*Copyright statement.* TEXT

## 1 Introduction

Low cloud cover continues to be a dominant source of uncertainty in projecting future climate (e.g. Bony and Dufresne, 2005; Dufresne and Bony, 2008; Vial et al., 2013), with variations in shallow cumulus distributions explaining much of the differences in climate model-derived estimates of climate sensitivity (e.g. Wyant et al., 2006; Medeiros and Stevens, 2011; Nam et al., 2012). This stems from climate models' inability to simulate shallow cumulus and their impacts, due in part to the low temporal and spatial resolution of these models (e.g. Stevens et al., 2002), as well as the fact that small-scale processes important for cloud development, including turbulence and convection, must be parameterized (e.g. Tiedtke, 1989; Zhang and McFarlane, 1995; Bretherton et al., 2004). Studies have shown precipitation efficiency is a key parameter used to constrain cloud parameterizations within climate models (Rennó et al., 1994; Del Genio et al., 2005; Zhao, 2014; Lutsko and Cronin,

2018). Nam et al. (2012) hypothesized that shallow cumulus are too reflective in climate models, possibly because model
precipitation efficiencies are too weak. This results in excess cloud water which increases cloud optical depth and shallow
cumulus reflectance. Prior observational and modeling studies found the precipitation efficiency of shallow cumulus increases
as sea-surface temperature (SST) increases in response to climate change (Lau and Wu, 2003; Bailey et al., 2015; Lutsko and
Cronin, 2018). Factors including environmental moisture (e.g. Heus and Jonker, 2008; Schmeissner et al., 2015), entrainment
(e.g. Korolev et al., 2016; Pinsky et al., 2016b, a), and aerosols (e.g. Koren et al., 2014; Dagan et al., 2016; Jung et al., 2016b, a)
help regulate both thermodynamic and dynamical processes that promote favorable conditions important to not only warm rain
production, but also the efficiency of the conversion of cloud water to precipitation. To better constrain cloud parameterizations
of these processes and subsequently climate sensitivity to low cloud cover, more observations-based studies analyzing physical
processes influencing warm rain efficiencies are needed.

In an ideal shallow cumulus cloud, liquid water content increases adiabatically from cloud base to top. However, liquid
water content is generally only 50% - 80% of the adiabatic values due to entrainment (Gerber et al., 2008; Blyth et al., 2013;
Watson et al., 2015). Evaporation induced by cloud-edge mixing not only impacts shallow cumulus updraft strength, but also
the number and size of droplets within a cloud (Lu et al., 2012), with increased evaporation potentially reducing the number
and size of available droplets. Using a large-eddy simulation (LES), Moser and Lasher-Trapp (2017) found the influence of
entrainment decreases from cloud-edge to center of individual shallow cumulus as they grow larger. This results in liquid water
content at cloud center being closer to adiabatic in larger clouds, because fewer droplets evaporate away at cloud-center. This
implies that the collision-coalescence process is more efficient at cloud center, because there is more cloud water available
to be collected by large droplets. As a result, smaller droplets originating near cloud-top may be more likely to continuously
grow larger as they fall, potentially reaching raindrop size near cloud base. At cloud edge, there are not only fewer droplets but
also smaller droplets, potentially reducing collision-coalescence efficiencies there. This is consistent with other LES results
that found shallow cumulus updrafts are more insulated from entrainment as they increase in size (e.g. Heus and Jonker, 2008;
Burnet and Brenguier, 2010; Tian and Kuang, 2016).

LES and limited field-campaign observational studies have shown that cloud updrafts not only become more protected as
cloud size increases, but also as relative humidity (RH) increases (e.g. Heus and Jonker, 2008; Schmeissner et al., 2015). Using
a model, Romps (2014) found precipitation efficiency to be closely related to RH, defining the lower bound of precipitation
efficiency as $\geq$ 1 - RH. Therefore, the precipitation efficiency at any given level of the atmosphere should increase with
increasing RH in response to lower evaporation rates. This suggests that lower RH would result in increased evaporation
rates and lower warm rain efficiencies. Prior studies have defined precipitation efficiency in two ways: 1) as the large-scale
precipitation efficiency and 2) as the cloud microphysical precipitation efficiency. Generally, observational studies have based
their definition of precipitation efficiency on the large-scale definition, which has simply been defined as the ratio of surface
rain rate to the sum of both vapor mass flux in/out of a cloud and surface evaporation (e.g. Chong and Hauser, 1989; Tao
et al., 2004; Sui et al., 2007), whereas the cloud microphyisical definition, or the ratio of surface rain rate to the sum of vapor
condensation and deposition rates, has been primarily used in cloud modeling studies (e.g. LI et al., 2002; Sui et al., 2005;
Gao et al., 2018). Although both the large-scale and cloud microphysical definitions of precipitation efficiency are useful (Sui

et al., 2005; Sui et al., 2007), variations in the ratio of cloud water to rain water (WRR) in response to changes in evaporation can theoretically be used as a proxy for warm rain efficiency based on the cloud microphysical definition. From this coupled with LES results showing that shallow cumulus updrafts are more protected as clouds grow in size and/or RH increases, *we hypothesize larger droplets will be evident closer to the cloud base and increase WRR in larger cloud objects, because the cloud-core of larger cloud objects is more protected from entrainment.*

While perhaps not as important as organization (Minor et al., 2011) or cloud size (Jiang and Feingold, 2006), it is widely understood that aerosol concentrations act to suppress warm rain production (Twomey, 1974; Albrecht, 1989) by increasing the cloud droplet concentration and reducing cloud droplet sizes (Squires, 1958). Albrecht (1989) found that increasing precipitation efficiency within a model is equivalent to decreasing the amount of cloud concentration nuclei (CCN), which reduces the mass concentration of cloud water within a cloudy layer. Similarly, Saleeby et al. (2015) used a cloud model to recently show that the number concentration of smaller cloud drops increases, but the number concentration of rain drops decrease as CCN increase in the presence of increasing aerosols. Lebsock et al. (2011) used CloudSat and Moderate Resolution Imaging Spectroradiometer (MODIS) observations to show that as drop size decreases, the ratio of rain water to cloud water also decreases. Together, these studies suggest the number of large droplets able to fall at sufficient terminal velocities to initiate collision-coalescence and continue growing to large enough sizes to fall out as rain decreases with increasing aerosol concentrations, which would reduce WRR.

Earlier studies have used satellite observations to infer the relationship between precipitation efficiency and both sea-surface temperature (Lau and Wu, 2003) and drop size (Lebsock et al., 2011). However, the relationship between cloud water and precipitation as shallow cumulus grow larger, environmental moisture increases, and/or as aerosol loading varies has only been investigated using cloud models (e.g. Abel and Shipway, 2007; vanZanten et al., 2011; Franklin, 2014; Saleeby et al., 2015; Moser and Lasher-Trapp, 2017; Hoffmann et al., 2017) and limited field-campaign observations (e.g. Rauber et al., 2007; Gerber et al., 2008; Burnet and Brenguier, 2010; Watson et al., 2015; Jung et al., 2016b). While these case and model studies provide insight into the physical processes, it is unclear how well they represent the shallow cumulus clouds observed globally. Satellites can observe a large enough sample size of shallow cumulus over different regions and during different stages of their lifecycle to gain a more holistic view of this relationship. Prior studies have used TRMM and Global Precipitation Measurement Mission (GPM) observations to analyze warm rain production and efficiency (e.g. Lau and Wu, 2003). Unfortunately, TRMM and GPM are precipitation radars operating at the Ku- and Ka-bands not capable of observing the non-raining portions of clouds or light precipitation. Building off work in Smalley and Rapp (2020) that analyzed the relationship between rain likelihood and cloud size, this study uses the higher sensitivity radar of CloudSat in addition to MODIS observations to test the hypothesis that *WRR is higher in larger shallow cumulus and is modulated by RH and aerosol loading*.

## 2  Data and Methods

To determine if larger shallow cumulus clouds are more efficient at producing warm rainfall, this study uses the CloudSat Cloud Profiling Radar (CPR; Tanelli et al., 2008) to identify individual contiguous shallow cumulus cloud objects. The CPR is

a near-nadir pointing 94-GHz radar that can observe raining and non-raining cloud drops. It allows us to analyze the horizontal distribution of cloud within a horizontal footprint of 1.4 x 1.8 km, and the vertical distributions of clouds within a 240 m bin within each cloudsat pixel.

Contiguous cloudy regions are initially identified using the 2B-GEOPROF (Marchand et al., 2008) cloud mask confidence values $\geq 20$, which removes orbit elements that may be influenced by ground clutter (Marchand et al., 2008). An additional limitation of CloudSat is it's inability to sense the smallest cloud droplets (e.g. Lamer et al., 2020). Smalley and Rapp (2020) addressed this by including CALIPSO measurements, which are sensitive to the smallest cloud droplets, in their identification of contiguous cloudy regions. However for this study, cloud objects must not be missing any reflectivity values. As a result,

some cloud object edges may not be the true edge, and some of our defined cloud objects may be connected to other cloud objects. Before identifying cloud objects, 2C-RAIN-PROFILE (Lebsock and L'Ecuyer, 2011) modeled reflectivity is mapped onto the two-dimensional cloud mask field. As outlined by prior literature (e.g. L'Ecuyer and Stephens, 2002; Mitrescu et al., 2010; Lebsock and L'Ecuyer, 2011), modeled reflectivity adjusts the raw reflectivity for multi-scattering and attenuation when it is raining. As described by Smalley and Rapp (2020), we use a lower-tropospheric stability threshold of 18.55 K (Klein

and Hartmann, 1993) to separate cloud objects occurring in environments favoring stratocumulus development from those occurring in environments favoring shallow cumulus development. To ensure that none of the cloud objects examined here contain ice, we only include cloud objects with tops entirely below the freezing level as defined in 2C-PRECIP-COLUMN (Haynes et al., 2009).

    Shallow cumulus cloud objects are then identified using the methodology described by Smalley and Rapp (2020) using the

combined two-dimensional reflectivity field, with only single-layer cloud objects included. We use the incidence precipitation flag from 2C-PRECIP-COLUMN (rain possible, probable, or certain) to identify raining cloud objects and the raining pixels within them. Using all three rain flags helps us identify pixels only producing light drizzle that might be evaporating before reaching the surface to those producing heavier rainfall (Haynes et al., 2009). This range of rainfall is incorporated into the integrated precipitation water path product from 2C-RAIN-PROFILE (Lebsock, 2018), and we use this product to determine

the average rain water path ($W_P$) for each cloud object, only including $W_P$ associated with raining pixels in the average. We then store the median cloud-top height and maximum along-track extent (hereby extent) of each cloud object for later analysis.

    Although CloudSat 2B-CWC-RVOD (Austin et al., 2009) does provide a cloud water path ($W_C$) product, the rain drop size distribution used in 2B-CWC-RVOD is not the same as that used in 2C-RAIN-PROFILE. Additionally, Christensen et al. (2013) found that the 2B-CWC-RVOD algorithm struggles to filter out precipitation sized droplets in the presence of light

precipitation and drizzle, which results in an overestimation of cloud water. This, coupled with differences in assumed drop size distributions by 2B-CWC-RVOD and 2C-RAIN-PROFILE, makes 2B-CWC-RVOD $W_C$ not ideal for this study, so we instead use MODIS $W_C$. Cho et al. (2015) found that MODIS effective radius and optical depth retrieval failure rates are higher in regions of broken trade cumulus than regions of predominantly stratocumulus, and they primarily attributed this to the presence of partially filled and inhomogeneous cloudy pixels. They also found that a large fraction of unexplained MODIS

retrieval failures are related to the presence of precipitation after comparing MODIS failure rates to non-precipitating and precipitating pixels classified by CloudSat. This is attributed to a higher frequency of failures due to effective radius being too

large. Considering the retrieval of effective radius and optical depth are required to derive $W_C$ and higher failure rates within broken trade cumulus, we suspect unavoidable sampling bias exists in $W_C$ matched to the smallest cloud objects and/or those containing large droplets and heavy rain. However on a global scale, prior studies have found the uncertainties in MODIS $W_C$

are small in comparison to other satellite retrievals (e.g. Seethala and Horvath, 2010; Lebsock and Su, 2014), with the global mean of MODIS $W_C$ being within 5 g m$^{-2}$ of $W_C$ determined using the Advanced Microwave Scanning Radiometer for Earth Observing System (AMSR-E) (Seethala and Horvath, 2010). Given potential uncertainties in $W_C$, we tested the sensitivity of our results to only including MODIS pixels with a minimum $W_C > 0$ g m$^{-2}$, 20 g m$^{-2}$, and 30 g m$^{-2}$ in our analysis, and we found that the overall interpretation of our results does not change depending on the minimum $W_C$ threshold used. Even though

our overall results do not change using a $W_C$ threshold below 30 g m$^{-2}$, we use the conservative estimate of $W_C$ ($\geq$ 30 g m$^{-2}$) which is based on an uncertainty estimate of 28 g m$^{-2}$ from Jolivet and Feijt (2005), coupled with an estimated uncertainty of 36 g m$^{-2}$ which was determined using error in effective radius and optical depth from Platnick and Valero (1995). Due to horizontal resolution differences between CloudSat and MODIS, one CloudSat pixel may overlap multiple MODIS pixels within a surrounding 3x3 km grid. As a result, $W_C$ is then calculated for each CloudSat pixel by averaging the nearest nine

non-zero MOD-06-1KM (Platnick et al., 2003) pixels within a 3x3 grid surrounding each CloudSat pixel, which have been previously matched to the CloudSat track in the MOD-06-1KM product (Cronk and Partain, 2018). There could be concerns that the averaging $W_C$ within the nearest nine MODIS pixels may not properly represent the $W_C$ at the appropriate scales relative to the horizontal footprint of each CloudSat pixel, however we tested our results using $W_C$ within the nearest MODIS pixel and found that our overall results do not change. We then store and analyze the mean $W_C$ associated with each cloud

object.

WRR of each shallow cumulus cloud object is calculated as $\frac{W_P}{W_C}$. Note, this is a proxy for true warm rain efficiency, because mass flux of water in and out of a cloud cannot be determined without a model; however, this ratio has been used by prior observational studies to analyze the amount of cloud water converted to rain water (e.g. Lebsock et al., 2011).

Considering Rayleigh reflectivity is a function of the drop size distribution to the sixth power, it is expected that the maximum

reflectivity in non-raining cloud objects will occur near cloud-top, then shift downward as a cloud transitions from non-raining to raining. Wang et al. (2017) used the vertical reflectivity gradient (VGZ) to investigate warm rain onset. They found VGZ (positive down) reverses sign (positive to negative) when clouds transition from non-raining to raining. Given previous studies and results shown in Smalley and Rapp (2020) finding rain is more likely as clouds grow larger in extent, it is hypothesized that the negative VGZ within individual raining cloud objects will increase in magnitude as cloud objects increase in extent. The

methodology developed by Wang et al. (2017) is applied to find the VGZ for each pixel within every shallow cumulus cloud object. VGZ at cloud object center pixel (VGZ$_{CP}$) will then be compared to VGZ at cloud object edge pixel (VGZ$_{EP}$) to infer the impact of mixing on cloud object cores as a function of cloud size and RH.

The influence of aerosols on the relationship between WRR and cloud object size are determined using Aqua MODIS level-3 daily 550 nm aerosol optical depth (AOD) (Ruiz-Arias et al., 2013). Each cloud object is matched to the nearest 1°x1° gridbox

AOD value. Note that AOD may not necessarily scale with the number of CCN due to its dependence on particle size, and that aerosol type varies globally. Additionally, AOD, being column integrated, does not give any information about where

the aerosols are within the atmospheric column, so high AOD does not necessarily mean that aerosols are occurring within the cloud layer. Finally, multiple studies have shown that AOD depends on relative humidity (Su et al., 2008; Michel Flores et al., 2012; Neubauer et al., 2017; Liu and Li, 2018). This results in aerosols swelling due to the uptake of water and an underestimation of the first indirect aerosol effect (Liu and Li, 2018). These conditions are not considered in this study but may factor into WRR.

As in Smalley and Rapp (2020), this analysis is constrained to only marine shallow cumulus between between 60 N and 60 S. Measurements are constricted to June 2006 and December 2010 because CloudSat stopped taking night time measurements after 2010 due to a battery anomaly (Witkowski et al., 2012). RH is classified using 6-hourly ECMWF-AUX (Cronk and Partain, 2017). However, because lateral mixing at shallow cumulus edges would most likely be entraining boundary layer air (see review by de Rooy et al., 2013), we tested the sensitivity of our results to RH at different pressure levels (850-mb and 950-mb) in the lower atmosphere, at the surface, and the average RH at or below 850-mb. We found that, while the magnitudes slightly change, the overall interpretation of our results does not depend on our definition of RH. As a result, we classify RH as the average RH at or below 850-mb and match it to each cloud object. Cloud-top height, RH, VGZ, and AOD are used to control and analyze the relationship between WRR and cloud object extent.

## 3   Warm rain relationship to extent

Similar to Smalley and Rapp (2020), The spatial distribution of $W_P$, $W_C$, WRR, AOD, and extent of raining shallow cumulus cloud objects is analyzed by binning them to a 2.5° x 2.5° global grid.

Figure 1a shows the spatial distribution of $W_P$ over the global ocean basins, with $W_P$ increasing equatorward. This is consistent with prior literature that found raining shallow cumulus are most frequent within the tropics (e.g. Smalley and Rapp, 2020). $W_P$ is largest near the Inter-Tropical Convergence Zone (ITCZ), South Pacific Convergence Zone (SPCZ), and tropical warm pool, with values exceeding 45 g m$^{-2}$. Deep convection is more frequent here (e.g. Waliser and Gautier, 1993), so some objects may be transitioning from raining shallow cumulus to deeper convection. The results likely include a mix of frequently occurring tropical raining shallow cumulus and the early stages of developing deep convection possibly resulting in large $W_P$ over the tropics.

Spatial patterns in $W_C$ (Figure 1b) within the tropics generally follow $W_P$, with values ranging between 110 g m$^{-2}$ and 150 g m$^{-2}$ in the tropics. We find that relative humidity generally decreases from median values near 90% in the tropics to median values near 80% north or south into the midlatitudes (not shown), this is consistent with modeling studies that found less cloud water evaporates away in wetter environments (e.g. Tian and Kuang, 2016). Considering boundary layer depth scales with SST (e.g. Wood and Bretherton, 2004b), the boundary layer is generally deeper over the tropical oceans than the sub-tropical oceans. This supports deeper clouds (e.g. Short and Nakamura, 2000; Rauber et al., 2007; Smalley and Rapp, 2020) and could also help explain why $W_C$ and WP are largest in the tropics.

Figure 1c shows the spatial patterns in WRR follow spatial patterns in $W_P$, with values increasing equatorward. Shallow cumulus cloud object WRR is largest within the ITCZ, SPCZ, and tropical warm pool, with values > 0.35. This is consistent

with Lau and Wu (2003), who found precipitation efficiency is positively correlated with SST (e.g. Lau and Wu, 2003), and implies that WRR is higher in wetter environments.

Patterns in spatial extent shown in Figure 1d are similar to those found by Smalley and Rapp (2020), who used combined CloudSat/CALIPSO to define extent, with extent decreasing from the stratocumulus regions west into the trade cumulus regions and north of the trade cumulus and stratocumulus regions into the ITCZ. Interestingly, Figure 1c shows WRR also peaks in the southeast Pacific stratocumulus region, implying that WRR is high in regions with relatively low SST. However, Figure 1e shows that fewer than 40 shallow cumulus objects are observed in a given gridbox over this region in a four-year period, reducing confidence in WRR here. Together, Figures 1c and 1d indicate that the relationship between WRR and extent is complicated and potentially depends on cloud depth (which increases in the tropics) and on environmental conditions including RH and aerosol loading.

To determine how WRR depends on cloud size, Figure 2 shows WRR as a function of cloud object extent. Note, we estimate the uncertainty in median WRR at any given extent by bootstrapping WRR at a given extent 10,000 times with replacement. Error in WRR median is then classified as $\pm$ one standard deviation of the bootstrapped sample distribution of median values. Similar error estimates are shown in Figures 3-5 later in this section. WRR follows a double power-law relationship, with WRR < 0.25 for cloud objects < 8.4 km and approaching 0.30 for cloud objects > 8.4 km. There is also very little spread in median WRR at a given extent which gives us confidence that this relationship is real. Similar to these results, earlier studies have shown a double power-law distribution in shallow cumulus size (e.g. Benner and Curry, 1998; Trivej and Stevens, 2010), which will be discussed in further detail later.

To address the impact of RH and cloud depth on WRR, Figure 3 shows the relationship between WRR and cloud object extent conditioned using cloud-top height and RH at or below 850-mb. Holding RH constant, WRR depends strongly on cloud-top height with WRR nearly doubling for each 0.5 km increase in cloud top height for a given extent in the most humid environments. For a given RH and top height, there is also an increase in WRR with extent. Holding top height constant, there is also an increase in WRR with increasing RH, with no overlap in median WRR error at a given extent or RH. However, increases in WRR are dominated by changing cloud size (depth and extent).

To support the hypothesis that larger shallow cumulus are able to sustain a larger droplet field within their cores to increase the precipitation efficiency, the variation in the VGZ across individual cloud objects is examined. We expect that VGZ will be a larger negative value near cloud center than cloud , especially as cloud size increases. As an example, Figure 4a shows the change in median $VGZ_{CP}$ to $VGZ_{EP}$ for cloud objects with an extent of 8.4 km. VGZ decreases from -3.48 dBZ km$^{-1}$ at cloud object edge to -20.3 dBZ km$^{-1}$ at cloud object center. Given the relationship between reflectivity and drop size, a negative $VGZ_{CP}$ implies that drop growth is occurring from near cloud top to near cloud base close to cloud object center, suggesting that larger droplets may be present near cloud base near cloud object center compared to the edge. To directly analyze drop size near cloud base, Figure 4b shows the spread in median near base reflectivity for cloud objects with an extent of 8.4 km. Figure 4b confirms that cloud drops are largest near cloud object center, with a median reflectivity of -5.28 dBZ. Reflectivity values, and subsequent drop sizes, then decrease moving from cloud object center to cloud object edge, with edge values of

-17.96 dBZ. Figure 4a coupled with 4b implies, at least for extents of 8.4 km, drops grow larger near cloud object centers and may be more protected from mixing.

Figure 4c shows the relationship between $VGZ_{CP}$ and $VGZ_{EP}$ as a function of extent and top height. For a constant cloud-top height, $VGZ_{CP}$ again follows a double power-law distribution. Specifically, the magnitude of the $VGZ_{CP}$ rapidly increases from approximately 10 dBZ km$^{-1}$ to 20 dBZ km$^{-1}$ as extent approaches 8.4 km, while it plateaus around 20 dBZ km$^{-1}$ for extents $> 8.4$ km. Conversely, $VGZ_{EP}$ decreases in magnitude, approaching 0 dBZ km$^{-1}$ for the largest cloud object extents. However, it does not decrease as fast as $VGZ_{CP}$, implying that the change in vertical reflectivity gradient in the center of cloud is driving changes in differences from center to edge. Figure 4c also shows that the change in $VGZ_{CP}$ depends on cloud-top height for extents $> 5.6$ km, with larger magnitudes for the tallest clouds. Narrowing this down to the possible influence of entrainment on cloud object updrafts from cloud edge to center, this is also consistent with previous modeling studies that found larger shallow cumulus cloud cores are more insulated from entrainment (e.g. Burnet and Brenguier, 2010; Tian and Kuang, 2016), a more adiabatic cloud core of developing cumulus as shown in Figure 2 from Moser and Lasher-Trapp (2017), and a higher probability of rainfall (e.g. Smalley and Rapp, 2020) in observations.

To determine how $VGZ_{CP}$ influences the relationship between WRR and extent, Figure 4d shows WRR as a function of extent conditioned by top height and $VGZ_{CP}$, with WRR increasing as the magnitude of $VGZ_{CP}$ increases; however, changes in WRR are not distinct when the magnitude of $VGZ_{CP}$ is larger than -15 dBZ km$^{-1}$ for extents $< 7$ km. This, coupled with Figure 4c, illustrates that as shallow cumulus grow deeper and wider, drops at the center of the cloud can grow larger and scavenge more available cloud water. This is consistent with larger shallow cumulus being more efficient at producing rainfall, perhaps in part because they are less influenced by environmental mixing.

Until this point, this paper has focused on how cloud size and RH impacts WRR. However, it is also understood that aerosol concentrations influence both the number and size of droplets within a cloud, with larger aerosol concentrations resulting in a greater number of smaller droplets (e.g. Twomey, 1974; Albrecht, 1989). As a result, we hypothesize increasing aerosol concentrations, which vary regionally (Figure 1f), increase the ratio of cloud droplets to rain drops, thus reducing WRR.

Figure 5a shows the relationship between WRR and AOD, conditioned by top height. On first glance, it appears that WRR increases as a function of AOD, which contradicts the expectation of a shift in drop size distribution towards fewer large drops to initiate collision-coalescence which would reduce the amount of cloud water converted to rain water. However, disentangling aerosol-cloud interactions from other meteorological variables is quite difficult, as increasing aerosol concentrations are often correlated with other environmental variables (e.g. Koren et al., 2014).

Given the strong dependence of WRR on top height, we further examine the relationship between AOD and top height (Figure 5b), conditioned by extent. The curves shown in Figure 5a look similar to those shown in Figure 5b, suggesting the positive correlation between aerosols and top height are responsible for the observed relationship between AOD and WRR. Indeed, Figure 5c further supports this assertion. When conditioned by top height, WRR shows little dependence on AOD, and suggests that the conversion from $W_C$ to $W_P$ is more sensitive to cloud depth than aerosols. While these results seem counterintuitive, this analysis examines clouds in which precipitation has been detected. Figure 5d shows the likelihood of rain occurrence at a given AOD determined by the ratio of raining cloud objects to the total number of cloud objects. As

expected, Figure 5d shows that the likelihood of rain decreases as AOD increases, with rain likelihood of about 50% in the cleanest environments decreasing to about 40% for an AOD approaching 0.75. These results imply that once the condensation-coalescence is initiated, aerosol loading has a smaller impact on the conversion of cloud water to rain than other cloud or environmental characteristics.

## 4    Limitations of analysis and observations

This study has emphasized the potential for the decreasing impact of entrainment on cloud cores, resulting in higher WRR, as cloud size increases; however, it is important to point out other factors related to cloud size that may also impact WRR. Figure 3 shows WRR is higher when cloud objects are taller, which may be simply because we are sampling more mature clouds that have had more time for the collision-coalescence process to result in rain formation. Deeper shallow cumulus not only live longer which would give cloud droplets more time to grow to raindrop size (e.g. Burnet and Brenguier, 2010), but they are more likely to have more intense updrafts which could result in more water vapor being transported to higher altitudes within a cloud. Stronger updrafts are then more likely to be able to suspend cloud droplets higher in the cloud for longer periods of time which allows them to grow larger before they begin to fall and collision-coalescence is initiated. Once cloud droplets do begin to fall, they are not only potentially larger but able to collect more droplets over a larger distance than droplets falling through a shallower cloud. This could potentially result in higher WRR, however there is likely a lag between the peaks in cloud water path and rain water path as cloud drops grow to raindrop size in a developing cloud. Earlier modeling studies have also noted that turbulent flow potentially enhances the likelihood of warm rain formation (e.g. Brenguier and Chaumat, 2001; Seifert et al., 2010; Wyszogrodzki et al., 2013; Franklin, 2014; Seifert and Onishi, 2016; Chen et al., 2018). Seifert et al. (2010) found that turbulence effects are largest near cloud tops in shallow cumulus, which they note is an important region for initial rain formation. While these additional processes may impact WRR, the satellite observations used in this study are instantaneous snapshots in time. We attempted to remove some of these life cycle impacts by binning cloud objects by top height. Within a given cloud top height bin, WRR (Figure 3) and the magnitude of $VGZ_{CP}$ (Figure 4c) still increase as a function of extent. While we acknowledge that this cannot fully remove these impacts, these results support the idea that processes other than those related to cloud lifetime, like lateral entrainment, may also influence the WRR of shallow cumulus of different horizontal sizes.

It is surprising that this study identifies shallow cumulus cloud objects larger than 10 km. This suggests that some stratocumulus are not being filtered out of this dataset by our LTS threshold. However, a majority of cloud objects that we identify have extents below 10 km. This is consistent with Figure 1e which shows that a majority of cloud objects occur over regions generally associated with shallow cumulus. To further test this, we performed the same analysis over the south pacific trade region but excluded the southeast stratocumulus region, and we still find few large cloud objects with our overall results and interpretation not changing. This suggests that predominant type of entrainment impacting these cloud objects would be lateral entrainment at cloud edges (see review by de Rooy et al., 2013), and that these are indeed shallow cumulus.

At the small end of the shallow cumulus horizontal size spectrum, CloudSat is limited to observing cloud objects no smaller than 1.4 x 1.8 km. Given prior ground observational studies, it is likely that there is a significant population of shallow cumulus cloud objects not identified by our study (e.g. Kollias et al., 2003; Mieslinger et al., 2019) due to non-uniform beam filling effects. Battaglia et al. (2020) noted that this results in an underestimation of path integrated attenuation, potentially introducing error into the retrieval of Wp. Unfortunately, this limitation is unavoidable given CloudSat's horizontal resolution.

## 5    Summary and Discussion

This study uses the methodology described by Smalley and Rapp (2020) to classify a large global shallow cumulus cloud object dataset from CloudSat and determine the relationship between WRR, cloud extent, RH, and aerosol loading. We find that WRR increases as a function of cloud size (top height and extent) and RH. Benner and Curry (1998) found a double-power law distribution in shallow cumulus thickness as a function of cloud diameter, and Trivej and Stevens (2010) hypothesized that the shift from one power-law distribution to another results from small shallow cumulus that can rapidly grow in size until reaching the trade inversion. We find a similar relationship between WRR and extent, showing that one distribution exists with WRR increasing faster for extents $< 8.4$ km then slowly increasing above this breakpoint. Trivej and Stevens (2010) also found that environmental factors, particularly RH, become important once cloud-top height reaches the trade inversion. Our results show that WRR is most sensitive to RH above an extent of 8.4 km, which we assume represents the average extent where cloud objects reach the trade inversion.

Unexpectedly, we find that for a fixed cloud depth, WRR is fairly insensitive to AOD. One explanation may be that, although high AOD values do occur over the global ocean basins, the majority of cloud objects being sampled still form in relatively clean air, so the minority of cloud objects occurring over polluted regions have a small impact on the overall statistics. Another explanation may be that this analysis only includes precipitating clouds, so once collision-coalescence is initiated, the amount of cloud water converted to rain water is less influenced by aerosol concentrations.

Past studies conclude that precipitation efficiency increases as SST increases (Lau and Wu, 2003; Bailey et al., 2015; Lutsko and Cronin, 2018). Considering warmer SSTs tend to result in deeper clouds (e.g. Wood and Bretherton, 2004a) and more humid environments (e.g. Chen and Liu, 2016), it is reasonable to expect that WRR would increase in response (e.g. Lau and Wu, 2003). Our results show that WRR is highest near the equator where SSTs are warmest. However, the general relationship between cloud size (depth and extent), RH, and WRR suggests that WRR is more sensitive to cloud size than RH. To directly address the SST dependence, Figure 6 shows the frequency distribution of extents and the median WRR, both as a function of cloud-top height and SST. For a given cloud-top height, WRR does increase as a function of SST. However, for a fixed SST, WRR also increases as extent increases. Additionally, Figure 6 shows that the frequency distribution of cloud object sizes shifts toward more frequent larger extents with increasing SST. Together, these suggest that increasing WRR with SST shown in past studies not only results from the deepening clouds but also the shift towards more frequent larger clouds.

Prior literature has shown that modeled shallow cumulus cores become more adiabatic as they grow larger (Moser and Lasher-Trapp, 2017), potentially resulting in larger drops. Figure 6 and our analysis of the relationship between VGZCP, extent,

and WRR suggest drop growth is being enhanced near the base at the center of larger cloud objects, potentially resulting in more cloud water being scavenged by larger droplets and more efficient autoconversion and accretion processes. Most climate models parameterize autoconversion and accretion as functions of cloud and precipitation properties (e.g. Lohmann and Roeckner, 1996; Liu and Daum, 2004; Morrison et al., 2005; Lim and Hong, 2010; Lee and Baik, 2017), but recently enhancement factors that depend on variations and covariations in WC and WP have been introduced to correct for biases due to subgrid-scale Wc and Wp inhomogeneity (e.g. Lebsock et al., 2013; Boutle et al., 2014; Witte et al., 2019). Presumably, the dependence of these enhancement factors on Wc variability would capture the increase in WRR with cloud depth shown here, however it is unclear if these enhancement factors based on the variance in Wc and Wp capture the effects of cloud extent on WC and WP, and subsequently WRR. Our dataset provides an opportunity for a future analysis that could focus on investigating the relationship between subgrid-scale variability in WC, WP, WRR, and extent, which could help improve our understanding and simulation of precipitating shallow cloud processes in climate models.

*Data availability.* All CloudSat/MODIS data products used in this analysis were acquired from the CloudSat Data Processing Center and can be accessed at http://www.cloudsat.cira.colostate.edu.

*Code and data availability.* Please contact the authors for access to any dataset created by the analysis and/or the code used to process the CloudSat/MODIS data..

*Author contributions.* Kevin Smalley performed the analysis. While, Kevin Smalley and Anita Rapp wrote and edited this manuscript.

*Competing interests.* The authors declare they have no conflicts of interest.

*Acknowledgements.* This research was supported by NASA grant NNX14AO72G.

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

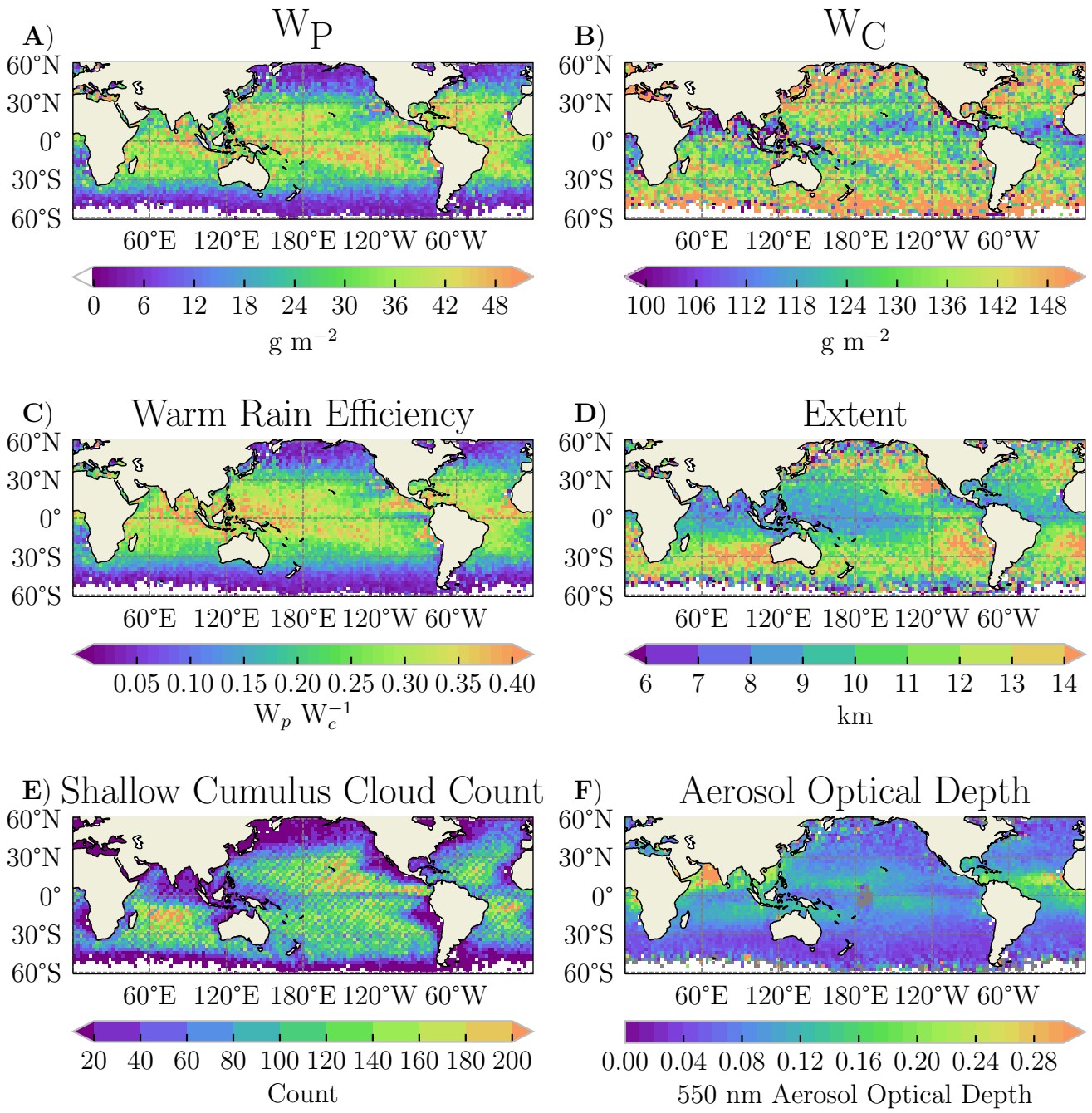

**Figure 1.** The spatial distribution of integrated precipitation water path ($W_P$), cloud water path ($W_C$), warm rain efficiency, extent, number of shallow cumulus cloud objects, and aerosol optical depth are shown in panels A), B), C), D), E), and F) respectively. Cloud objects are binned onto a 2.5° x 2.5° spatial grid, and any grid box containing no data is white.

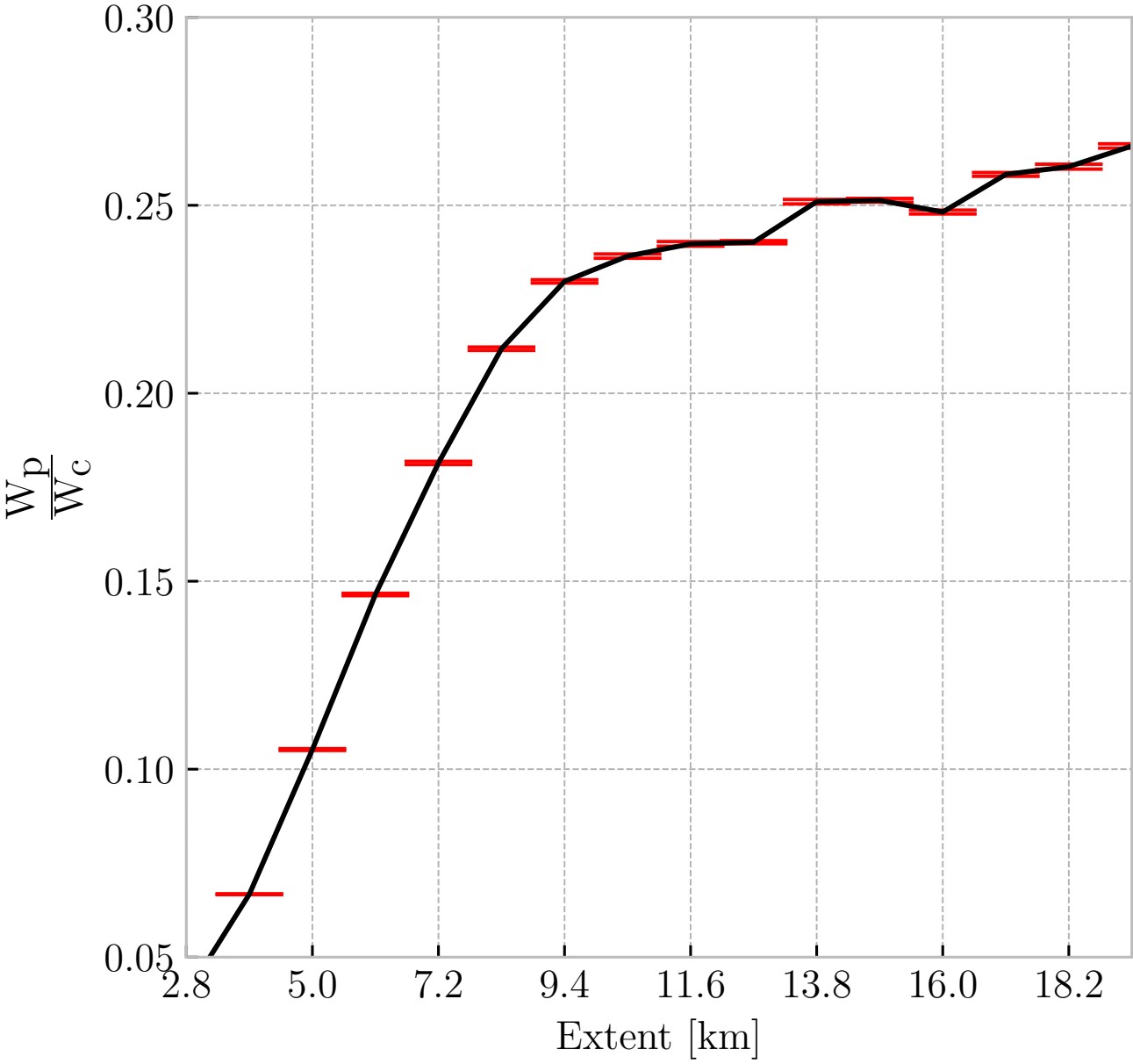

**Figure 2.** The median ratio of cloud water to rain water $\left(\frac{W_p}{W_c}\right)$ at a given maximum size (extent). The red errorbars represent $\pm 1$ standard deviation of a bootstrapped distribution of $\left(\frac{W_p}{W_c}\right)$ medians at a given extent.

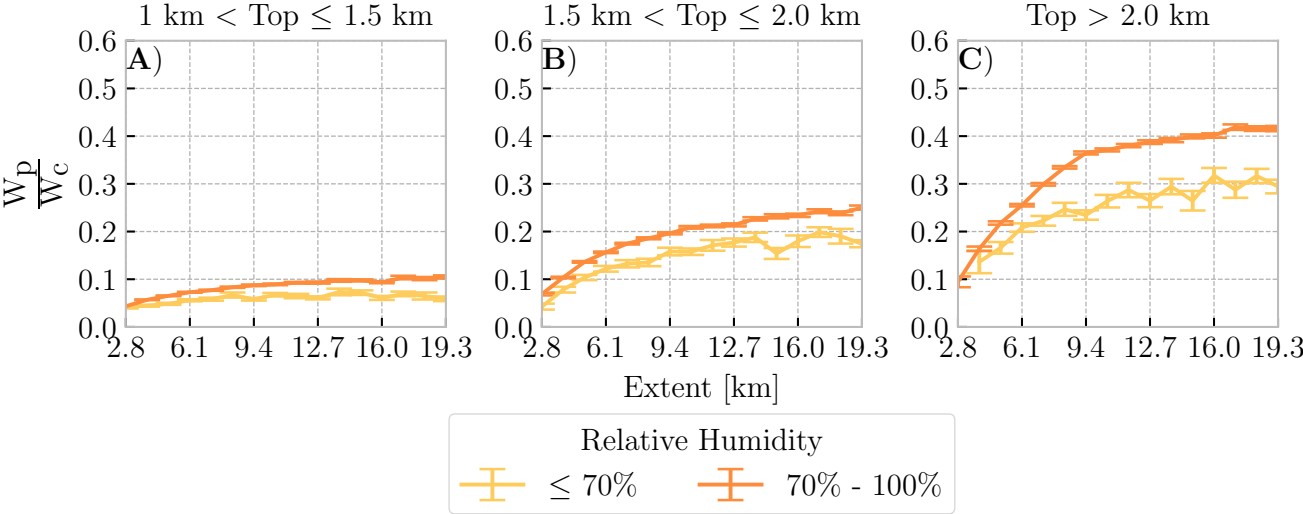

**Figure 3.** The median ratio of cloud water to rain water $\left(\frac{W_p}{W_c}\right)$ at a given maximum size (extent). The different line colors represent cloud objects separated by < 850-mb relative humidity (RH). Errorbars represent $\pm 1$ standard deviation of a bootstrapped distribution of $\left(\frac{W_p}{W_c}\right)$ medians at a given extent and RH.

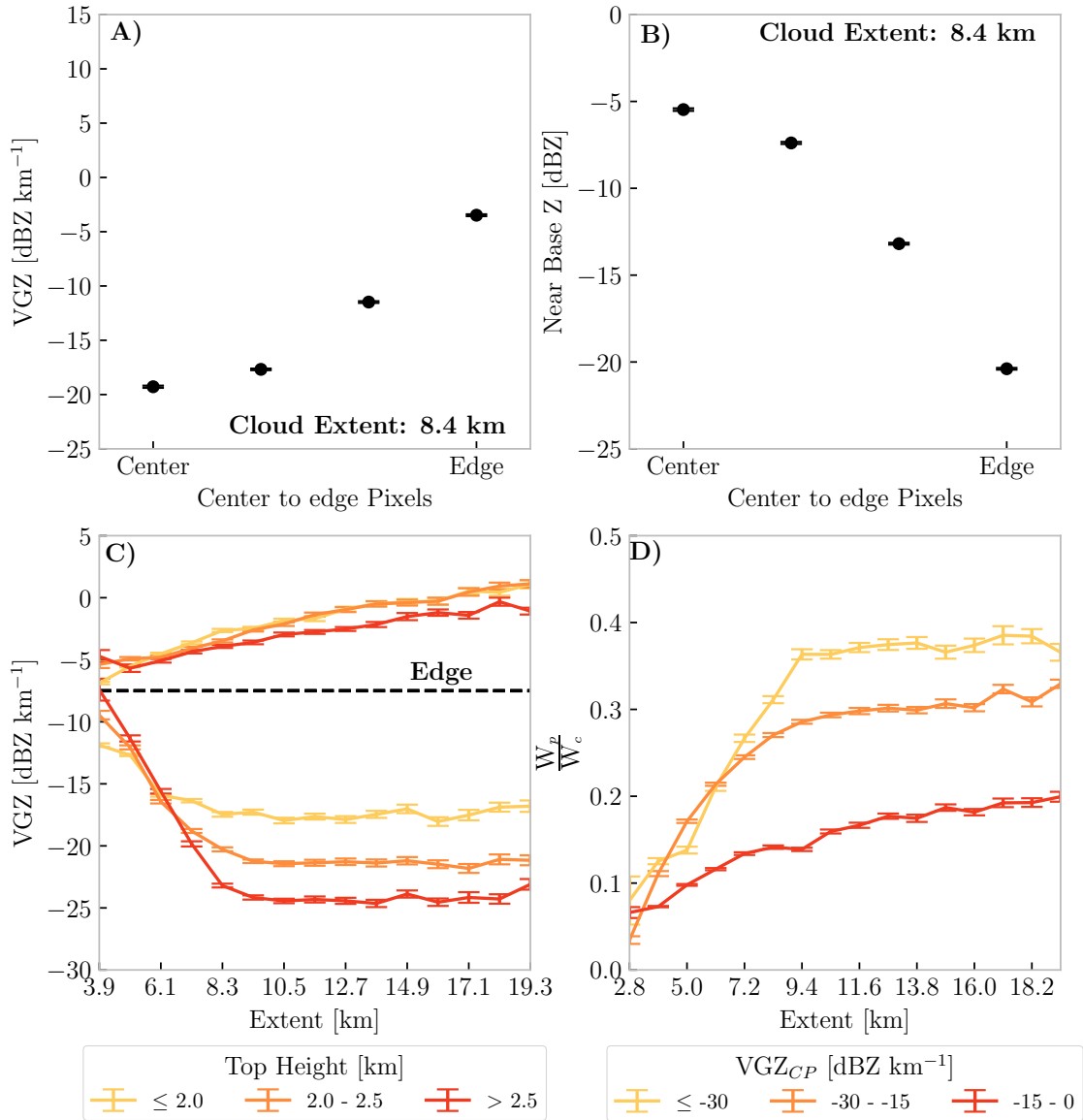

**Figure 4.** Panel A) shows the median change in the vertical reflectivity (VGZ) from the center to edge of all cloud objects with an extent of 8.4 km. Panel B) shows the median change in near base reflectivity (Z) from the center to edge of all cloud objects with an extent of 8.4 km. Panel C) shows the median vertical reflectivity gradient (VGZ) at the center and edge of different sized (extent) raining cloud objects. Different lines represent cloud objects separated by top height. Panel D) shows the median ratio of cloud water to rain water ($\frac{W_P}{W_C}$) at a given median size (extent). The different line colors represent cloud objects separated by the vertical reflectivity gradient on the center pixel (VGZ$_{cp}$) of all cloud objects. Error bars represent $\pm 1$ standard deviation of a bootstrapped distribution of median VGZ and Z for a given pixel from cloud object edge to center (Panels A) and B)), as well as VGZ and $\frac{W_P}{W_C}$ at a given extent (Panels C) and D)).

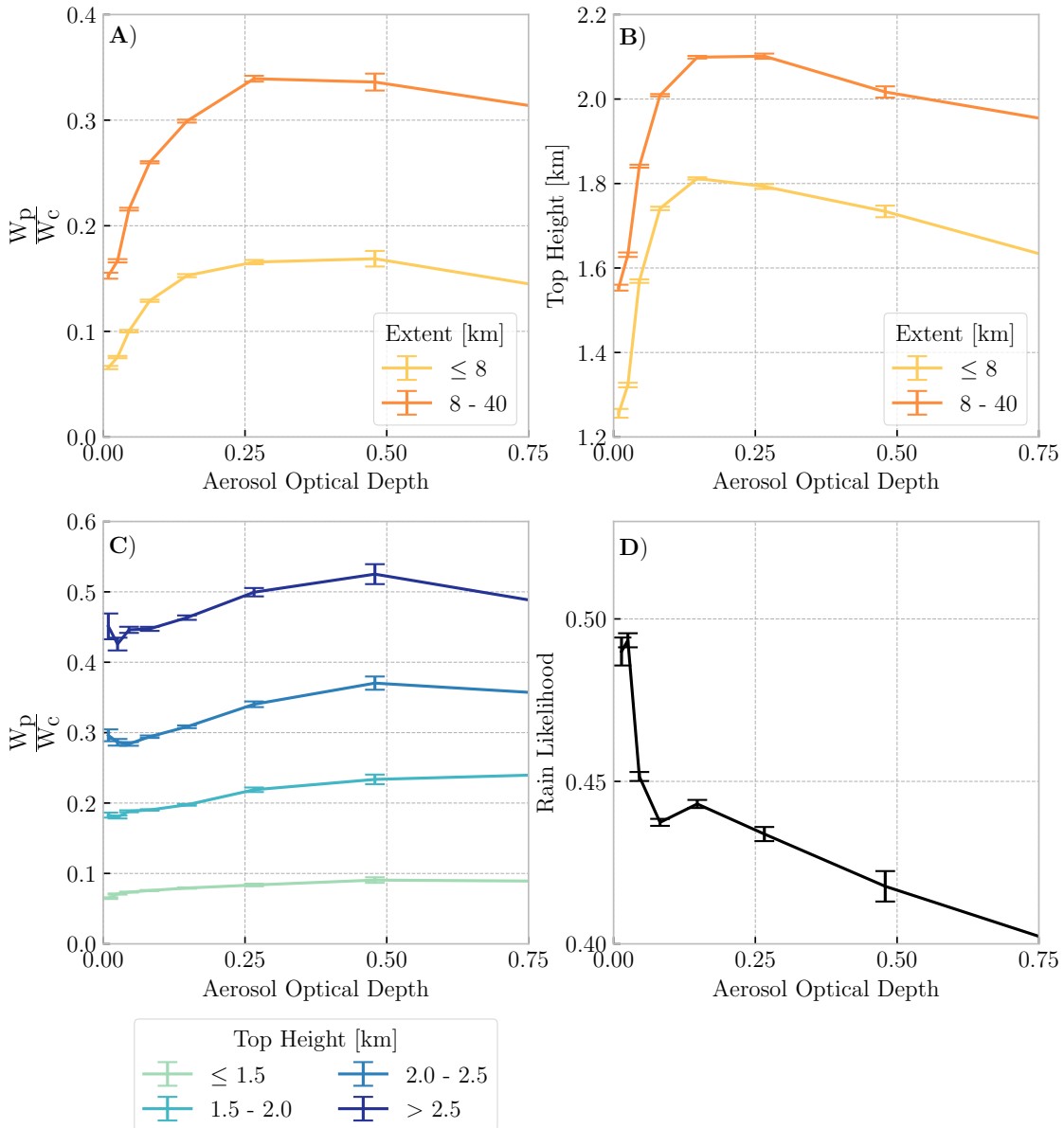

**Figure 5.** Panel A) shows the relationship between median warm rain efficiency as MODIS 550 nm aerosol optical depth. Panel B) shows the relationship between median cloud-top height and aerosol optical depth. Panel C) shows the relationship between warm rain efficiency $\left(\frac{W_p}{W_c}\right)$ and aerosol optical depth. Line colors in panels A) and B) represent cloud objects separated by extent, while line colors in panel C) represent cloud objects separated by top height. Panel D) shows the ratio of raining cloud objects to non-raining cloud objects (rain likelihood) at a given aerosol optical depth. For panels A), B), and C), errorbars represent ±1 standard deviation of a bootstrapped distribution of raining cloud objects to determine the uncertainty in $\frac{W_p}{W_c}$, and top height at a given aerosol optical depth. Whereas, the errorbars shown in panel D) represent ±1 standard deviation of a bootstrapped distribution of raining and non-raining cloud objects to determine rain likelihood uncertainty at a given aerosol optical depth.

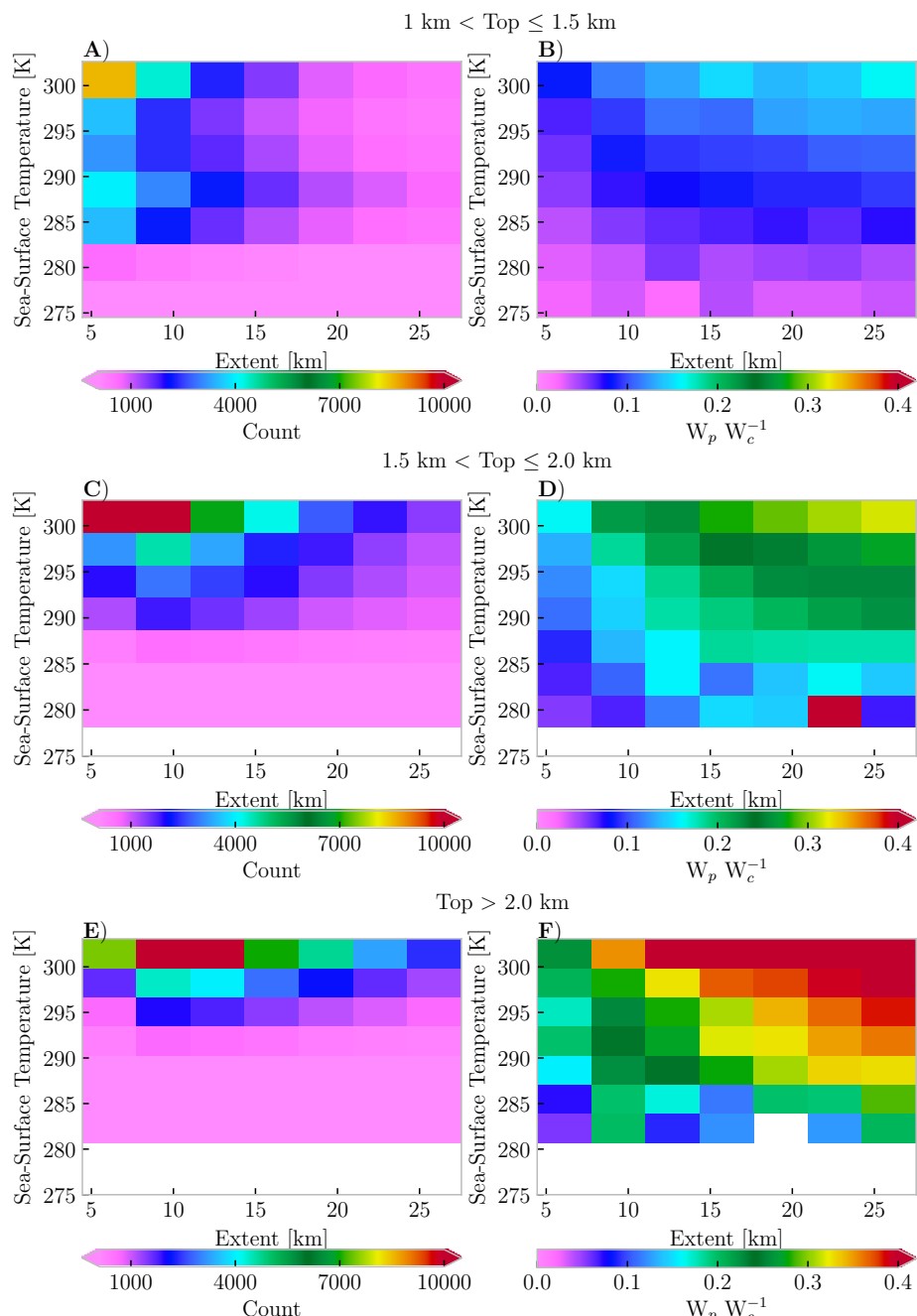

**Figure 6.** The two-dimensional distribution of extent as a function of sea-surface temperature, conditioned by cloud-top height, is shown in panels A), C), and E) respectively. The median ratio of cloud water to rain water ($W_p \ W_c^{-1}$) as a function of Extent and sea-surface temperature are shown in panels B), D), and F) respectively.