# Peer review of "A-Train estimates of the sensitivity of the cloud to rain water ratio to cloud size, relative humidity, and aerosols"

_Atmospheric Chemistry and Physics, 2020_

## Referee Comment (RC1) · Anonymous Referee #1 · 1 Oct 2020

Overview

The authors examine warm rain efficiency (WRE) in marine liquid clouds using rain water path estimates from the CloudSat Cloud Profiling Radar and cloud water path from MODIS. They show that WRE increases as cloud extent increases after controlling for cloud top height and low level relative humidity. AOD shows little correlation with WRE when conditioned by cloud top height indicating potentially limited aerosol impacts on WRE once warm rain has begun. WRE increases as expected with SST due to clouds that are deeper with more condensed water but this study also shows that WRE also increases with cloud extent for a given SST and cloud extents grow with SST. Thus,

increased WRE as SST increases could also partly result from larger clouds that are more protected from dry air entrainment.

I can somewhat buy into the primary argument of this study that larger clouds are more protected from deleterious dry air entrainment and thus are more apt to form rain. However, there are a number of additional considerations that have to be discussed before such a conclusion can be reached, highlighted in the major comments below. In addition, some plots and methods used need improvement, again highlighted in more detail below.

Major Comments

1. The title indicates that "warm rain likelihood" is examined in addition to warm rain efficiency but nearly the entire study focuses on warm rain efficiency and does not consider clouds that are not already raining. Thus, I recommend changing the title of the paper.

2. The clouds being analyzed in this study are repeatedly referred to as shallow cumulus clouds but the cloud length scales examined are 1.7 to 18 km, so this is combining quite robust cumulus clouds at the short end of the spectrum with wider presumably stratocumulus clouds at the longer end of the spectrum. This makes the "shallow cumulus" terminology misleading for me. Applying a simple 18.55 K lower tropospheric stability separator will not filter out all stratocumulus clouds. In addition, many shallow cumulus are smaller than the CloudSat CPR footprint of 1.8 x 1.4 km, so the clouds at this end of the spectrum will suffer from non-uniform beam filling that can bias retrievals (e.g., Battaglia et al. 2020).

3. The assumption that clouds are shallow cumulus feeds into the assumption that lateral entrainment is the key process controlling WRE, which is stated repeatedly throughout the study. Lateral entrainment is important for km-scale cumulus clouds but cloud top entrainment is important for 10s of kilometers scale stratocumulus clouds. In addition, is there anything to suggest that once a relatively shallow liquid cloud is

wider than multiple kilometers that its core is not protected from lateral entrainment? There are too many assumptions being made regarding the importance of entrainment without supporting evidence.

Another potentially major contributor to warm rain efficiency that also correlates with cloud size is cloud lifetime, which should be discussed but isn't. Larger clouds typically live longer, which could increase the probability of rain formation. Other factors that could impact WRE that are not mentioned but should be include turbulent enhancement of droplet collision-coalescence, updraft speed controls on the supersaturation and number of droplets condensed, and potential time lags between peaks in rain water path and cloud water path due to raindrops consuming cloud droplets.

4. Lines 67-68: It is made to seem like there are very few studies examining relationships between cloud water and precipitation in shallow cumulus as a function of cloud size, moisture, or aerosol conditions, but this isn't true, and I encourage a more through literature review. For example, consider the many studies that have been published using Dominica Experiment (DOMEX) field campaign data. A number of field campaigns and modeling studies have focused on entrainment and precipitation formation in cumulus clouds over land and ocean, and even more have examined stratocumulus clouds. 5. The methodology could use some improvements and clarifications. a. Line 80: The CloudSat CPR cannot always observe non-raining cloud drops because its sensitivity is limited, which has been proven with comparisons to ground sensors (Lamer et al. 2020). In addition, it has ground clutter issues below 1-km altitude. These are important caveats that should be mentioned that could bias sampling. b. What are the uncertainties of the rain water path and cloud water path estimates? On line 92, it indicates that any rain water paths greater than 0 are considered but there should be a minimum value used that is equal to the retrieval uncertainty. For example, for cloud water path, this is typically $\sim$20 g m-2. c. Lines 122-123: Average relative humidity below 3 km is a very strange metric for environmental moisture when most of these shallow clouds are interacting with a variable altitude inversion layer. This metric would mix boundary layer air with typically much drier free tropospheric air, which would be weighted by the inversion altitude (which increases as one moves from stratocumulus to trade cumulus regions). The relevant moisture metric for lateral or cloud top entrainment would be the relative humidity in the lower free troposphere.

6. The single line in Figure 2 begs for the spread to be shown and statistical significance tests to be performed. The same applies to Figures 3-5. How large is the spread? Are the median lines shown statistical significant? In addition, some numbers and symbols are missing in the legends of Figure 3-5. Lastly, edge lines in Figure 4b are not blue as described in the caption.

7. Lines 117-119: More important caveats to list than the type of aerosol not being considered are AOD not necessarily scaling with CCN number due to its dependence on size, AOD being offset from the actual clouds, AOD being column integrated such that aerosols may not be making it into the cloud, and AOD being positively correlated with relative humidity due to aerosol swelling.

8. The studies cited on lines 176-177 as supporting the conclusion that more protection from entrainment is what is causing the larger clouds to rain more are not necessarily relevant in that they are analyzing kilometer-scale cumulus congestus and deep convective clouds, not 10 km wide shallow clouds.

Minor Comments

1. Lines 47-50: Romps (2014) examined precipitation efficiency with respect to relative humidity but relative humidity typically remains approximately constant over oceans as a function of temperature and it is absolute humidity that increases with SST and temperature, so Lau and Wu (2003) is not consistent with Romps (2014) because one is analyzing relative humidity, which impacts evaporation rate, while the other is examining absolute humidity, which impacts condensed mass.

2. Lines 50-53: Why are larger droplets necessarily expected near cloud base? Drizzle

typically forms first near the top of the cloud in an updraft where the condensed mass and turbulence is greatest. Is it the falling of this drizzle and collection of cloud droplets during falling that produces the largest droplets near cloud base?

3. Line 58: Please clarify whether cloud water and raindrop concentration refer to number concentration or mass concentration.

4. Line 66: missing a verb after "aerosol loading".

5. Line 103: Symbol is missing in parentheses.

6. Line 107: Insert "Rayleigh" before "reflectivity".

7. Lines 135-138: More important than relative humidity impacted evaporation to increasing rain water path is absolute humidity, which controls how much condensation occurs.

8. Lines 146-147: Is "east" supposed to be "west"? And why is "north" used with respect to the ITCZ?

9. Line 160: Be more specific than "environmental moisture". This implies absolute humidity but in fact what is analyzed is relative humidity.

10. Lines 165-168: The different vertical gradients of reflectivity near cloud edges as compared to near cloud centers does not conclusively show that larger droplets are present near cloud base at cloud center than on the edge because we don't know the absolute reflectivity magnitudes.

References

Battaglia, A., Kollias, P., Dhillon, R., Lamer, K., Khairoutdinov, M., and Watters, D., 2020: Mind the gap – Part 2: Improving quantitative estimates of cloud and rain water path in oceanic warm rain using spaceborne radars, Atmos. Meas. Tech., 13, 4865–4883, https://doi.org/10.5194/amt-13-4865-2020.

Lamer, K., Kollias, P., Battaglia, A., and Preval, S., 2020: Mind the gap – Part 1: Accurately locating warm marine boundary layer clouds and precipitation using spaceborne radars, Atmos. Meas. Tech., 13, 2363–2379, https://doi.org/10.5194/amt-13-2363-2020.
* * *

---

## Referee Comment (RC2) · Anonymous Referee #2 · 6 Oct 2020

This paper is a useful analysis of the production of warm rain in cumulus clouds based primarily on cloud and rain water measurements from the CloudSat and MODIS satellite datasets. The main new result is that the efficiency of production of warm rain appears to increase with the horizontal size of the cloud, even when controlling for variations in cloud depth and sea surface temperature. The results imply that dilution of cloud updrafts due to entrainment is less effective in larger clouds than smaller clouds which are presumably better protected by the larger scale of the clouds. This is a plausible hypothesis supported by some prior modeling. The paper shows consistent results between an examination of the ratio of precipitation water to cloud water and the vertical gradient in CloudSat reflectivity. I have some comments about the resolution of the measurements used, the quantification of "warm rain efficiency", and the conclusions the authors draw about the aerosol sensitivity of warm rain efficiency. The paper should be suitable for publication in ACP subject to some revisions.

Some aspects of the scales of the clouds in this investigation are left unanswered, but are potentially critical because of the resolution of the measurements employed. The CloudSat rain water data used here has a footprint of 1.4 x 1.8 km. The cloud water path data from MODIS has a nominal resolution of ~1 km at nadir. According to the methods, the cloud water path is based on a 9-pixel average, which suggests that the horizontal scale of the cloud water measurements are on the scale of 10 km. Nevertheless, clouds are shown varying from about 1.7 km to greater than 18 km. So, one question is: are the cloud water values really representative of the true values for clouds smaller than 10 km? Can we then be certain that the strong dependence of the ratio of precipitation water to cloud water on cloud scale shown in figure 2 for clouds smaller than 10 km is not influenced by the resolution of the cloud water quantity?

The authors state that "prior studies [of biases in MODIS cloud water] have found them to be small in comparison to other satellite retrievals". I suspect that this result may be resolution dependent and that in fact uncertainties for cloud smaller than several km in scale may be quite significant. For example, Cho et al. (2015) find that the MODIS cloud property retrievals from which the cloud water path is derived can have substantial errors in cumulus cloud fields because of partially cloudy pixels and horizontal inhomogeneity of cloud properties within the satellite footprint. Can the authors provide some greater support for the notion that the cloud water values are representative of the true value at the scales on the small end of the spectrum shown in this analysis?

Fine resolution satellite imagery indicates that warm cumulus clouds substantially smaller than 1.7 km are common and in fact may be more prevalent than clouds larger than 1.7 km (e.g. Mieslinger et al. 2019). Presumably some of these clouds may be precipitating. Obviously, comparable data to the CloudSat data are not available at smaller scales from satellite. Nevertheless, do the authors expect that there may be

a substantial population of precipitating cumulus clouds that are not captured in their analysis? Furthermore, one might expect that warm cumulus clouds might be limited in scale. Assuming crudely that cumulus clouds typically have an aspect ratio of around 1, one might presume that cumulus clouds broader than 5-10 km might also be tall enough to contain ice or mixed phase microphysical processes occurring. What characteristics ensure that the clouds included here are both warm liquid phase and truly cumulus clouds, or is the analysis expecting to include some stratocumulus clouds as well?

The authors use the ratio of precipitation water to cloud water as their measure of "warm rain efficiency". Although, as the authors note, this quantity is just a proxy for the true efficiency. I think the authors are correct to make this point clear. I also think that perhaps it would be helpful for the authors to clarify what defines a proper quantitative measure of the warm rain efficiency. Presumably, it is not so easily observed, which is why they have chosen a proxy, which is fine. Given the brevity of this paper, however, I think a short elaboration on this point would be helpful. Furthermore, if the ratio used in this paper is merely a proxy for the true efficiency, is it really appropriate to be using "warm rain efficiency" throughout the manuscript to refer to this quantity? I suggest that the authors perhaps consider a different name so that readers are not confused about what is the true measure of the efficiency and what is the approximation of it. Alternatively, if there is a quantitative comparison of the ratio to the true efficiency, perhaps from a theoretical study, then it might be appropriate to refer to the proxy value as a measure of the efficiency with some quoted uncertainty value.

The corroboration of the inferences based on the ratio of precipitating water to cloud water with the inferences from the vertical gradient in reflectivity (VGZ) is a valuable contribution of this paper and certainly strengthens the case that the authors are making. In lines 174 to 180 the authors argue that the dependence of VGZ on cloud-top height supports the notion that updrafts in larger clouds are protected from entrainment. Why would this dependence on cloud-top height not simply result from collision/coalescence happening through a deeper cloud layer independent of any difference in entrainment? Presumably the taller clouds are provide a broader distance from cloud base to cloud top through which raining drops can fall and collect cloud drops. Likewise, perhaps a stronger updraft that yields a taller cloud is better at promoting the coalescence of cloud drops through turbulent collisions. Could these similarly explain the differences between clouds of differing heights?

Finally, the authors explore the dependence of their proxy for warm rain efficiency on the aerosol optical thickness in the vicinity of the cloud. They conclude that there is little dependence of the efficiency on aerosols, which is an interesting result. I suggest, though, that the authors remove the word "surprisingly" from the abstract where this result is reported. As noted by the authors, by excluding non-precipitating clouds from their analysis they are likely missing the expected dominant effect, which is the suppression of rain formation. Is there not a CloudSat study looking at the dependence of the occurrence of rain in CloudSat retrievals upon AOD? I think that a citation to such a study would be appropriate in the discussion of the results presented in this paper. If not, I think the authors should point out that this might be the more fruitful path to quantifying aerosol effects.

References: Cho, H.M., Zhang, Z., Meyer, K., Lebsock, M., Platnick, S., Ackerman, A.S., Di Girolamo, L., C.‐Labonnote, L., Cornet, C., Riedi, J. and Holz, R.E., 2015. Frequency and causes of failed MODIS cloud property retrievals for liquid phase clouds over global oceans. Journal of Geophysical Research: Atmospheres, 120(9), pp.4132-4154.

Mieslinger, T., Horváth, Á., Buehler, S.A. and Sakradzija, M., 2019. The dependence of shallow cumulus macrophysical properties on large‐scale meteorology as observed in ASTER imagery. Journal of Geophysical Research: Atmospheres, 124(21), pp.11477-11505.
* * *
[Figure]

2020.

---

## Author Comment (AC1) · 4 Dec 2020

We thank the reviewer for all their helpful critiques and suggestions that helped us improve this manuscript. Reviewer comments are given in black. Our responses are given in red, and the updated text in this document is blue.

The authors examine warm rain efficiency (WRE) in marine liquid clouds using rain water path estimates from the CloudSat Cloud Profiling Radar and cloud water path from MODIS. They show that WRE increases as cloud extent increases after controlling for cloud top height and low level relative humidity. AOD shows little correlation with WRE when conditioned by cloud top height indicating potentially limited aerosol impacts on WRE once warm rain has begun. WRE increases as expected with SST due to clouds that are deeper with more condensed water but this study also shows that WRE also increases with cloud extent for a given SST and cloud extents grow with SST. Thus, increased WRE as SST increases could also partly result from larger clouds that are more protected from dry air entrainment.

I can somewhat buy into the primary argument of this study that larger clouds are more protected from deleterious dry air entrainment and thus are more apt to form rain. However, there are a number of additional considerations that have to be discussed before such a conclusion can be reached, highlighted in the major comments below. In Addition, some plots and methods used need improvement, again highlighted in more detail below.

Major Comments

1. The title indicates that "warm rain likelihood" is examined in addition to warm rain efficiency but nearly the entire study focuses on warm rain efficiency and does not consider clouds that are not already raining. Thus, I recommend changing the title of the paper.
   a. **The title has been updated to "A-Train estimates of the sensitivity of the cloud to rain water ratio to cloud size, relative humidity, and aerosols".**

2. The clouds being analyzed in this study are repeatedly referred to as shallow cumulus clouds but the cloud length scales examined are 1.7 to 18 km, so this is combining quite robust cumulus clouds at the short end of the spectrum with wider presumably stratocumulus clouds at the longer end of the spectrum. This makes the "shallow cumulus" terminology misleading for me. Applying a simple 18.55 K lower tropospheric stability separator will not filter out all stratocumulus clouds. In addition, many shallow cumulus are smaller than the CloudSat CPR footprint of 1.8 x 1.4 km, so the clouds at this end of the spectrum will suffer from non-uniform beam filling that can bias retrievals (e.g., Battaglia et al. 2020).
   a. **The 18.55K LTS threshold has been commonly used in the literature as a robust separator between the two regimes (Klein and Hartmann, 1993). However, to further examine the possible influence of stratocumulus on our results, we re-ran the analysis for both the global oceans, excluding the southeast Pacific, northeast Pacific, southeast Atlantic, northeast Atlantic, and Indian ocean stratocumulus region. We also separately analyzed only the south Pacific trade cumulus region (excluding the southeast Pacific stratocumulus region). Our overall results and their interpretation do not change if we only analyze regions of mostly shallow cumulus. This lends credibility to LTS being an effective separator between shallow and stratocumulus, and that the majority of cloud objects we identify are shallow cumulus. However, given this may be a concern that future readers might have, we have added the following text (Pages 9-10, Lines 288-294) to the paper :** *"It is surprising that this study identifies shallow cumulus cloud objects larger than 10 km. This suggests that some stratocumulus are not being filtered out of this dataset by our LTS threshold. However, a majority of cloud objects that we identify have extents below 10 km. This is consistent with Figure 1e which shows that a majority of cloud objects occur over regions generally associated with shallow cumulus. To further test this, we performed the same analysis over the south pacific*

*trade region but excluded the southeast stratocumulus region, and we still find few large cloud objects with our overall results not changing. This suggests that predominant type of entrainment impacting these cloud objects would be lateral entrainment at cloud edges (see review by de Rooy et al., 2013), and that these are indeed shallow cumulus."*.

b. **With regards to your second point that there are likely shallow cumulus smaller than the CloudSat footprint, unfortunately this is unavoidable given CloudSat's resolution. We had discussed the potential impacts of beam filling and clouds below the satellite FOV side in our previous paper (Smalley and Rapp, 2020) that this one is a follow-on to, but did not repeat the discussion here. However, to address issues related to resolution and cloud scales, we have added the following text (Page 10, Lines 295-299) to the paper:** *"At the small end of the shallow cumulus horizontal size spectrum, CloudSat is limited to observing cloud objects no smaller than 1.4 x 1.8 km. Given prior ground observational studies, it is likely that there is a significant population of shallow cumulus cloud objects not identified by our study (e.g. Kollias et al., 2003; Mieslinger et al., 2019) due to non-uniform beam filling effects. Battaglia et al. (2020) noted that this results in an underestimation of path integrated attenuation, potentially introducing error into the retrieval of $W_p$. Unfortunately, this limitation is unavoidable given CloudSat's horizontal resolution."*.

3. The assumption that clouds are shallow cumulus feeds into the assumption that lateral entrainment is the key process controlling WRE, which is stated repeatedly throughout the study. Lateral entrainment is important for km-scale cumulus clouds but cloud top entrainment is important for 10s of kilometers scale stratocumulus clouds. In addition, is there anything to suggest that once a relatively shallow liquid cloud is wider than multiple kilometers that its core is not protected from lateral entrainment? There are too many assumptions being made regarding the importance of entrainment without supporting evidence. Another potentially major contributor to warm rain efficiency that also correlates with cloud size is cloud lifetime, which should be discussed but isn't. Larger clouds typically live longer, which could increase the probability of rain formation. Other factors that could impact WRE that are not mentioned but should be include turbulent enhancement of droplet collision-coalescence, updraft speed controls on the supersaturation and number of droplets condensed, and potential time lags between peaks in rain water path and cloud water path due to raindrops consuming cloud droplets.

   a. **As described in major comment 2, we believe that we are sampling predominantly shallow cumulus. Excluding regions of the globe where stratocumulus are common does not change our overall findings. As to your other concerns regarding other possible factors contributing to changes in warm rain efficiency, we agree that there are potential processes other than entrainment which may contribute to higher warm rain efficiency with cloud size, and they are now described in the following text (Page 9, Lines 268-287):** *"This study has emphasized the potential for the decreasing impact of entrainment on cloud cores, resulting in higher WRR, as cloud size increases; however, it is important to point out other factors related to cloud size that may also impact WRR. Figure 3 shows WRR is higher when cloud objects are taller, which may be simply because we are sampling more mature clouds that have had more time for the collision-coalescence process to result in rain formation. Deeper shallow cumulus not only live longer which would give cloud droplets more time to grow to raindrop size (e.g. Burnet and Brenguier, 2010), but they are more likely to have more intense updrafts which could result in more water vapor being transported to higher altitudes within a cloud. Stronger updrafts are then more likely to be able to suspend cloud droplets higher in the cloud for longer periods of time which allows them to grow larger before they begin to fall and collision-coalescence is initiated. Once cloud droplets do begin to fall, they are not only potentially larger but able*

*to collect more droplets over a larger distance than droplets falling through a shallower cloud. This could potentially result in higher WRR, however there is likely a lag between the peaks in cloud water path and rain water path as cloud drops grow to raindrop size in a developing cloud. Earlier modeling studies have also noted that turbulent flow potentially enhances the likelihood of warm rain formation (e.g. Brenguier and Chaumat, 2001; Seifert et al., 2010; Wyszogrodzki et al., 2013; Franklin, 2014; Seifert and Onishi, 2016; Chen et al., 2018). Seifert et al. (2010) found that turbulence effects are largest near cloud tops in shallow cumulus, which they note is an important region for initial rain formation. While these additional processes may impact WRR, the satellite observations used in this study are instantaneous snapshots in time. We attempted to remove some of these life cycle impacts by binning cloud objects by top height. Within a given cloud top height bin, WRR (Figure 3) and the magnitude of $VGZ_{CP}$ (Figure 4c) still increase as a function of extent. While we acknowledge that this cannot fully remove these impacts, these results support the idea that processes other than those related to cloud lifetime, like lateral entrainment, may also influence the WRR of shallow cumulus of different horizontal sizes".*

4. Lines 67-68: It is made to seem like there are very few studies examining relationships between cloud water and precipitation in shallow cumulus as a function of cloudsize, moisture, or aerosol conditions, but this isn't true, and I encourage a more thorough literature review. For example, consider the many studies that have been published using Dominica Experiment (DOMEX) field campaign data. A number of field campaigns and modeling studies have focused on entrainment and precipitation formation of cumulus clouds over land and ocean, and even more have examined stratocumulus clouds.

   a. **In reference to the sentence that you highlight, we have added more citations in support of those features being looked at primarily using cloud models and field-campaign observations. Please see the following additions to the text (Page 3, Lines 75-79):** *"However, the relationship between cloud water and precipitation as shallow cumulus grow larger, environmental moisture increases, and/or as aerosol loading varies has only been investigated using cloud models (e.g. Abel and Shipway, 2007; vanZanten et al., 2011; Franklin, 2014; Saleeby et al., 2015; Moser and Lasher-Trapp, 2017; Hoffmann et al., 2017) and limited field-campaign observations (e.g. Rauber et al., 2007; Gerber et al., 2008; Burnet and Brenguier, 2010; Watson et al., 2015; Jung et al., 2016b)."*.

5. The methodology could use some improvements and clarifications. a. Line 80: The CloudSat CPR cannot always observe non-raining cloud drops because its sensitivity is limited, which has been proven with comparisons to ground sensors (Lamer et al. 2020). In addition, it has ground clutter issues below 1-km altitude.These are important caveats that should be mentioned that could bias sampling. b.What are the uncertainties of the rain water path and cloud water path estimates? On line 92, it indicates that any rain water paths greater than 0 are considered but there should be a minimum value used that is equal to the retrieval uncertainty. For example, for cloud water path, this is typically~20 g m-2. c. Lines 122-123: Average relative humidity below 3 km is a very strange metric for environmental moisture when most of these shallow clouds are interacting with a variable altitude inversion layer. This metric would mix boundary layer air with typically much drier free tropospheric air, which would be weighted by the inversion altitude (which increases as one moves from stratocumulus to trade cumulus regions). The relevant moisture metric for lateral or cloud-top entrainment would be the relative humidity in the lower free troposphere.

a.  The cloud mask threshold of greater than or equal to 20 from
    2B-GEOPROF was chosen because it confidently removes CloudSat
    pixels that may be influenced by ground clutter. We describe this on
    Page 4, Lines 94-95, *"Contiguous cloudy regions are initially
    identified using the 2B-GEOPROF (Marchand et al., 2008) cloud mask
    confidence values ≥ 20, which removes orbit elements that may be
    influenced by ground clutter (Marchand et al., 2008)."*. Additionally,
    the following clarification *"An additional limitation of CloudSat is it's
    inability to sense the smallest cloud droplets (e.g. Lamer et al., 2020).
    Smalley and Rapp (2020) addressed this by including CALIPSO
    measurements, which are sensitive to the smallest cloud droplets, in
    their identification of contiguous cloudy regions. However for this
    study, cloud objects must not be missing any reflectivity values. As
    a result, some cloud object edges may not be the true edge, and
    some of our defined cloud objects may be connected to other cloud
    objects."* has been added on Page 4, Lines 95-100 to address the
    caveat regarding CloudSat not being sensitive to the smallest
    non-raining cloud drops.

b. In general, the uncertainty in rain water path varies by pixel depending on path integrated attenuation, uncertainty in cloud water, the drop size distribution, and evaporation. To address this comment, we used the incidence precipitation flag from 2C-PRECIP-COLUMN (Haynes et al. 2009). Specifically, we use the least strict definition of raining pixels (rain possible, probable, and certain) to identify raining pixels within cloud objects, because using the most strict definition of raining pixels (rain certain) biases our results to only larger cloud objects and removes cloud objects that are likely producing light drizzle which are identified using both rain possible and probable. Additionally, Lebsock et al. (2011) used the same three flags to identify raining pixels in their analysis of cloud water to rain water for similar reasons. As a result, we believe using all three rain incidence flags to identify raining pixels and match $W_p$ to each cloud object. For specifics, see the following text (Page 4, Lines 108-114) which has been added to the paper: "*We use the incidence precipitation flag from 2C-PRECIP-COLUMN (rain possible, probable, or certain) to identify raining cloud objects and the raining pixels within them. Using all thre rain flags helps us identify pixels only producing light drizzle that might be evaporating before reaching the surface to those producing heavier rainfall (Haynes et al., 2009). This range of rainfall is incorporated into the integrated precipitation water path product from 2C-RAIN-PROFILE (Lebsock, 2018), and we use this product to determine the average rain water path ($W_p$) for each cloud object, only including W P associated with raining pixels in the average.*".

c. As for cloud water path, we tested the sensitivity of the ratio of cloud water path to rain water path using pixels with no cloud water path threshold, a threshold of 20 $g\ m^{-2}$, and a threshold of 30 $g\ m^{-2}$ in our calculation of mean cloud object cloud water path. We chose 20 $g\ m^{-2}$ because it was suggested by the reviewer, and 30 $g\ m^{-2}$ as a conservative estimate based on an uncertainty estimate of 28 $g\ m^{-2}$ from Jolivet and Feijt (2005), and an uncertainty estimate of 36 $g\ m^{-2}$ using uncertainties in effective radius and optical thickness from Platnick and Valero (1995). We found that our results do not change based on the cloud water path uncertainty threshold that we use, therefore, based on studies mentioned above, we now only use MODIS pixels with cloud water path > 30 $g\ m^{-2}$ in our calculation of the ratio of cloud water path to rain water path in this study. See the following text (Page 5, Lines 131-136) that is now in the paper:

*"Given potential uncertainties in $W_C$ , we tested the sensitivity of our results to only including MODIS pixels with a minimum $W_C > 0\ g\ m^{-2}$ , 20 $g\ m^{-2}$ , and 30 $g\ m^{-2}$ in our analysis, and we found that the overall interpretation of our results does not change depending on the minimum $W_C$ threshold used. Even though our overall results do not change using a $W_C$ threshold below 30 $g\ m^{-2}$ , we use the conservative estimate of $W_C$ ($\geq$ 30 $g\ m^{-2}$ ) which is based on an uncertainty estimate of 28 $g\ m^{-2}$ from Jolivet and Feijt (2005), coupled with an estimated uncertainty of 36 $g\ m^{-2}$ which was determined using error in effective radius and optical depth from Platnick and Valero (1995)."*.

d. **The reason we chose below 3 km relative humidity as our moisture metric is because it was suggested and used in Smalley and Rapp (2020). However, we agree with this reviewer that boundary layer depth will not always be at or above 3 km. For a potentially better representation of boundary layer relative humidity, we tested the sensitivity of our results to relative humidity closer to the surface (850-mb, 925-mb, surface, and average below 850-mb). We found that the interpretation of our results were insensitive to the specific atmospheric level within the boundary layer we use to classify relative humidity. Therefore, we now use average below 850-mb relative humidity, which corresponds to a standard height of 1500 m, as a proxy for boundary-layer relative humidity. For specifics, please see the following text (Page 6, Lines 168-173) which has been added to the paper:** *"RH is classified using 6-hourly ECMWF-AUX (Cronk and Partain, 2017). However, because lateral mixing at shallow cumulus edges would most likely be entraining boundary layer air (see review by de Rooy et al., 2013), we tested the sensitivity of our results to RH at different pressure levels (850-mb and 950-mb) in the lower atmosphere, at the surface, and the average RH at or below 850-mb; We found that, while the magnitudes slightly change, the overall interpretation of our results does not depend on our definition of RH. As a result, we classify RH as the average RH at or below 850-mb and match it to each cloud object."*.

6. The single line in Figure 2 begs for the spread to be shown and statistical significance tests to be performed.[a] The same applies to Figures 3-5.[a] How large is the spread?[a] Are the median lines shown statistically significant?[a] In addition, some numbers and symbols are missing in the legends of Figure 3-5.[b] Lastly, edge lines in Figure 4 are not blue as described in the caption.[c]

   a. **We now use a monte carlo methodology to estimate the spread in sample median values at a given x value on each figure. We classify error in the median lines as plus/minus one standard deviation surrounding each median value at a given x value on each figure, considering lines significantly different if their associated error bars do not overlap. See the text for details on how we estimate error (Page 7, Lines 204-207):** *"Note, we estimate the uncertainty in median WRR at any given extent by bootstrapping WRR at a given extent 10,000 times with replacement. Error in WRR median is then classified as ± one standard deviation of the bootstrapped sample distribution of median values. Similar error estimates are shown in Figures 3-5 later in this section."*.
   b. **See the updated legends in Figures 3-5.**
   c. **See updated Figure 4 for correction.**

7. Lines 117-119: More important caveats to list than the type of aerosol not being considered are AOD not necessarily scaling with CCN number due to its dependence on size, AOD being offset from the actual clouds, AOD being column integrated such that aerosols may not be making it into the cloud, and AOD being positively correlated with relative humidity due to aerosol swelling.

   a. **These are definitely important caveats that must be discussed before using AOD to classify the influence of aerosols on warm rain. As a result, we added the following text (Pages 5-6, Lines 159-165) to the paper that accounts for these caveats:** *"Note that AOD may not necessarily scale with the number of CCN due to its dependence on particle size, and that aerosol type varies globally. Additionally, AOD, being column integrated, does not give any information about where the aerosols are within the atmospheric column, so high AOD does not necessarily mean that aerosols are occurring within the cloud layer. Finally, multiple studies have shown that AOD depends on relative humidity (Su et al., 2008; Michel Flores et al., 2012; Neubauer et al., 2017; Liu and Li, 2018). This results in aerosols swelling due to the uptake of water and an underestimation of the first indirect aerosol effect (Liu and Li, 2018). These conditions are not considered in this study but may factor into WRR."*.

8. The studies cited on lines 176-177 as supporting the conclusion that more protection from entrainment is what is causing the larger clouds to rain more are not necessarily relevant in that they are analyzing kilometer-scale cumulus congestus and deep convective clouds, not 10 km wide shallow clouds.

   a. **We removed the reference to Hernandez-Deckers and Sherwood (2018) and replaced it with a reference to Tian and Kuang (2016) which is more applicable to shallow cumulus, and modified the reference to Moser and Lasher-Trapp (2017) for clarity. See the following text (Page 8, lines 236-240) for changes** *"Narrowing this down to the possible influence of entrainment on cloud object updrafts from cloud edge to center, this is also consistent with previous modeling studies that found larger shallow cumulus cloud cores are more insulated from entrainment (e.g. Burnet and Brenguier, 2010; Tian and Kuang, 2016), a more adiabatic cloud core of developing cumulus as shown in Figure 2 from Moser and Lasher-Trapp (e.g. 2017), and a higher probability of rainfall (e.g. Smalley and Rapp, 2020) in observations."*.

Minor Comments

1. Lines 47-50: Romps (2014) examined precipitation efficiency with respect to relative humidity but relative humidity typically remains approximately constant over oceans as a function of temperature and it is absolute humidity that increases with SST and temperature, so Lau and Wu (2003) is not consistent with Romps (2014) because one is analyzing relative humidity, which impacts evaporation rate, while the other is examining absolute humidity, which impacts condensed mass.

   a. **Considering we wanted to highlight the potential influences of entrainment on warm rain efficiency, and that would be related to evaporation rates, we have removed the reference to Lau and Wu (2003) as you can see in the updated text (Page 2, Lines 48-51):** *"Using a model, Romps (2014) found precipitation efficiency to be closely related to RH, defining the lower bound of precipitation efficiency as ≥ 1 - RH. Therefore, the  precipitation efficiency at any given level of the atmosphere should increase with increasing RH in response to lower evaporation rates. This suggests that lower RH would result in increased evaporation rates and lower warm rain efficiencies."*.

2. Lines 50-53: Why are larger droplets necessarily expected near cloud base? Drizzle typically forms first near the top of the cloud in an updraft where the condensed massand turbulence is greatest. Is it the falling of this drizzle and collection of cloud droplets during falling that produces the largest droplets near cloud base?

   a. **The expectation is that a more efficient collision-coalescence process at cloud center will result in larger droplets, because the smaller droplets originating at the top of the cloud will fall through cloudy air with a higher amount of cloud water available for drop growth resulting in the largest drops near cloud base (See page 2, Line 42-43 for clarification):** *"As a result, smaller droplets originating near cloud-top may be more likely to continuously grow larger as they fall, potentially reaching raindrop size near cloud base."*.

3. Line 58: Please clarify whether cloud water and raindrop concentration refer to number concentration or mass concentration.

   a. **The reference to Albrect (1989) refers to cloud water mass concentration, while the reference to Saleeby et al. (2015) refers to raindrop number concentration. We clarified this in the following text (Page 3, Lines 65-69) in the paper:** *"Albrecht (1989) found that increasing precipitation efficiency within a model is equivalent to decreasing the amount of cloud concentration nuclei (CCN), which reduces the mass concentration of cloud water within a cloudy layer. Similarly, Saleeby et al. (2015) used a cloud model to recently show*

   b. *that the number concentration of smaller cloud drops increases, but the number concentration of rain drops decrease as CCN increase in the presence of increasing aerosols."*.

4. Line 66: missing a verb after "aerosol loading".

   a. **See the following text (Page 3, Line 76) for this correction:** *"However, the relationship between cloud water and precipitation as shallow cumulus grow larger, environmental moisture increases, and/or as aerosol loading varies"*.

5. Line 103: Symbol is missing in parentheses.

   a. **That should have been a reference to (Cronk and Partain, 2018), and see the following text (Page 5, Line 140) for this correction:** *"As a result, W C is then calculated for each CloudSat pixel by averaging the nearest nine non-zero MOD-06-1KM (Platnick et al., 2003) pixels within a 3x3 grid surrounding each CloudSat pixel, which have been previously matched to the CloudSat track in the MOD-06-1KM product (Cronk and Partain, 2018)."*. .

6. Line 107: Insert "Rayleigh" before "reflectivity".

    a. **As is shown in the following text (Page 5, Line 148) now in the paper, we now refer to "reflectivity" as "Rayleigh reflectivity" when it is first discussed in the methods:** *"Considering Rayleigh reflectivity is a function of the drop size distribution to the sixth power, it is expected that the maximum reflectivity in non-raining cloud objects will occur near cloud-top, then shift downward as a cloud transitions from non-raining to raining."*.

7. Lines 135-138: More important than relative humidity impacted evaporation to increasing rain water path is absolute humidity, which controls how much condensation occurs.

    a. **While it is true that absolute humidity is important to the amount of condensation that occurs, We find that relative humidity generally decreases from a median value of approximately 90% in the tropics to a median value of 80% as you move north or south towards the midlatitudes. Considering the large-scale environment (as defined using ECMWF) is generally not saturated, we would argue that relative humidity is the more important metric to reference here because there won't be any condensation if the environment does not reach saturation. To make this clear, we have modified the following text (Page 6, Lines 186-187) in the paper** *"We find that relative humidity generally decreases from median values near 90% in the tropics to median values near 80% north or south into the midlatitudes (not shown), this is consistent with modeling studies that found less cloud water evaporates away in wetter environments (e.g. Tian and Kuang, 2016)."*.

8. Lines 146-147: Is "east" supposed to be "west"? And why is "north" used with respect to the ITCZ?

    a. **It should say that extent decreases to the west from the stratocumulus regions into the trade cumulus regions Additionally, north is being used with respect to the ITCZ to say that the shallow cumulus cloud objects classified by Smalley and Rapp (2020) are also smaller in horizontal size (extent) north of both the trade cumulus and stratocumulus regions within the ITCZ region. To better clarify both of these points, , see the following text (page 7, lines 196-198) that has been modified in the paper:** *"Patterns in spatial extent shown in Figure 1d are similar to those found by Smalley and Rapp (2020), who used combined CloudSat/CALIPSO to define extent, with extent decreasing from the stratocumulus regions west into the trade cumulus regions and north of the trade cumulus and stratocumulus regions into the ITCZ."*.

9. Line 160: Be more specific than "environmental moisture". This implies absolute humidity but in fact what is analyzed is relative humidity.

   a. **We changed instances of "environmental moisture" in the abstract, results, and conclusions to "RH" (average RH at or below 850-mb).**

10. Lines 165-168: The different vertical gradients of reflectivity near cloud edges as compared to near cloud centers does not conclusively show that larger droplets are present near cloud base at cloud center than on the edge because we don't know the absolute reflectivity magnitudes.

    a. **We added a panel to Figure 4 (now Figure 4b) to show how reflectivity values near cloud base change from cloud object center to cloud object edge, and added the following text to pages 7-8, lines 224-229:** *"Figure 4b confirms that cloud drops are largest near cloud object center, with a median reflectivity of -5.28 dBZ. Reflectivity values, and subsequent drop sizes, then decrease moving from cloud object center to cloud object edge, with edge values of -17.96 dBZ. Figure 4a coupled with 4b implies, at least for extents of 8.4 km, drops grow larger near cloud object centers and may be more protected from mixing."*.

**References:**

Klein, S. A. and Hartmann, D. L.: The Seasonal Cycle of Low Stratiform Clouds, Journal of Climate, 6, 1587–1606, https://doi.org/10.1175/1520-0442(1993)006<1587:TSCOLS>2.0.CO;2, https://doi.org/10.1175/1520-0442(1993)006<1587:TSCOLS> 2.0.CO;2, 1993.

Smalley, K. M. and Rapp, A. D.: The role of cloud size and environmental moisture in shallow cumulus precipitation, Journal of Applied Meteorology and Climatology, 0, null, https://doi.org/10.1175/JAMC-D-19-0145.1, https://doi.org/10.1175/JAMC-D-19-0145.1, 2020

Haynes, J. M., LEcuyer, T. S., Stephens, G. L., Miller, S. D., Mitrescu, C., Wood, N. B., and Tanelli, S.: Rainfall retrieval over the ocean with spaceborne W-band radar, Journal of Geophysical Research, 114, https://doi.org/10.1029/2008jd009973, https://doi.org/10.1029% 2F2008jd009973, 2009.

Lebsock, M. D., L'Ecuyer, T. S., and Stephens, G. L.: Detecting the Ratio of Rain and Cloud Water in Low-Latitude Shallow Marine 465 Clouds, Journal of Applied Meteorology and Climatology, 50, 419–432, https://doi.org/10.1175/2010JAMC2494.1, https://doi.org/10. 1175/2010JAMC2494.1, 2011.

Jolivet, D. and Feijt, A. J.: Quantification of the accuracy of liquid water path fields derived from NOAA 16 advanced very high resolution radiometer over three ground stations using microwave radiometers, Journal of Geophysical Research: Atmospheres, 110,

https://doi.org/https://doi.org/10.1029/2004JD005205,
https://agupubs.onlinelibrary.wiley.com/doi/abs/10.1029/2004JD005205, 2005.

Platnick, S. and Valero, F. P. J.: A Validation of a Satellite Cloud Retrieval during ASTEX, Journal of the
Atmospheric Sciences, 52, 2985–3001,
https://doi.org/10.1175/1520-0469(1995)052<2985:AVOASC>2.0.CO;2, https://doi.org/10.1175/
1520-0469(1995)052<2985:AVOASC>2.0.CO;2, 1995.

Tian, Y. and Kuang, Z.: Dependence of entrainment in shallow cumulus convection on vertical velocity
and distance to cloud edge, Geophysical Research Letters, 43, 4056–4065,
https://doi.org/10.1002/2016gl069005, https://doi.org/10.1002%2F2016gl069005, 2016.

Hernandez-Deckers, D. and Sherwood, S. C.: On the Role of Entrainment in the Fate of Cumulus
Thermals, Journal of the Atmospheric
Sciences, 75, 3911–3924, https://doi.org/10.1175/jas-d-18-0077.1,
https://doi.org/10.1175%2Fjas-d-18-0077.1, 2018.

Moser, D. H. and Lasher-Trapp, S.: The Influence of Successive Thermals on Entrainment and Dilution in
a Simulated Cumulus Congestus, Journal of the Atmospheric Sciences, 74, 375–392,
https://doi.org/10.1175/JAS-D-16-0144.1, https://doi.org/10.1175/JAS-D-16-0144.1, 2017.

Lau, K. M. and Wu, H. T.: Warm rain processes over tropical oceans and climate implications,
Geophysical Research Letters, 30, https://doi.org/10.1029/2003GL018567,
https://agupubs.onlinelibrary.wiley.com/doi/abs/10.1029/2003GL018567, 2003.

Albrecht, B. A.: Aerosols, Cloud Microphysics, and Fractional Cloudiness, Science, 245, 1227–1230,
https://doi.org/10.1126/science.245.4923.1227,
https://science.sciencemag.org/content/245/4923/1227, 1989.

Saleeby, S. M., Herbener, S. R., van den Heever, S. C., and L'Ecuyer, T.: Impacts of Cloud
Droplet–Nucleating Aerosols on Shallow Tropical Convection, Journal of the Atmospheric
Sciences, 72, 1369–1385, https://doi.org/10.1175/JAS-D-14-0153.1, https://doi.org/10.1175/
JAS-D-14-0153.1, 2015.
Cronk, H. and Partain, P.: CloudSat ECMWF-AUX Auxillary Data Product Process Description and
Interface Control Document, Tech. rep., Colorado State University, 2017.

---

## Author Comment (AC2) · 4 Dec 2020

We appreciate the reviewer's helpful comments and suggestions that helped us improve this manuscript. Reviewer comments are in black. Our responses are red, and the updated text shown in this document is blue.

This paper is a useful analysis of the production of warm rain in cumulus clouds based primarily on cloud and rain water measurements from the CloudSat and MODIS satellite datasets. The main new result is that the efficiency of production of warm rain appears to increase with the horizontal size of the cloud, even when controlling for variations in cloud depth and sea surface temperature. The results imply that dilution of cloud updrafts due to entrainment is less effective in larger clouds than smaller clouds which are presumably better protected by the larger scale of the clouds. This is a plausible hypothesis supported by some prior modeling. The paper shows consistent results between an examination of the ratio of precipitation water to cloud water andthe vertical gradient in CloudSat reflectivity. I have some comments about the resolution of the measurements used, the quantification of "warm rain efficiency", and the conclusions the authors draw about the aerosol sensitivity of warm rain efficiency. The paper should be suitable for publication in ACP subject to some revisions.

Major Comments

1. Some aspects of the scales of the clouds in this investigation are left unanswered, but are potentially critical because of the resolution of the measurements employed. The CloudSat rain water data used here has a footprint of 1.4 x 1.8 km. The cloudwater path data from MODIS has a nominal resolution of~1 km at nadir. According To the methods, the cloud water path is based on a 9-pixel average, which suggests that the horizontal scale of the cloud water measurements are on the scale of 10 km. Nevertheless, clouds are shown varying from about 1.7 km to greater than 18 km. So,one question is: are the cloud water values really representative of the true values for clouds smaller than 10 km? Can we then be certain that the strong dependence of the ratio of precipitation water to cloud water on cloud scale shown in figure 2 for clouds smaller than 10 km is not influenced by the resolution of the cloud water quantity?

a. **To address the reviewer's concerns, we used a 3x3 grid (nine pixel) average surrounding each CloudSat pixel and only averaged cloud water path values that are > 0 g m$^{-2}$. We did this, because one CloudSat Pixel could overlap multiple MODIS pixels within that 3x3 grid, meaning that an average of multiple pixels is the best way to match MODIS cloud water path to each CloudSat pixel. However, we tested our results to check if matching the nearest pixel or nearest nine pixels would impact our results. We found that our results are consistent no matter what method is used to match MODIS cloud water path. To address this, the following text has been modified in the methods to clarify how we match cloud water path to each CloudSat pixel and mention that matching both a nine-pixel average and nearest-neighbor cloud water path does not change our overall results (Page 5, Lines 136-143)** *"Due to horizontal resolution differences between CloudSat and MODIS, one CloudSat pixel may overlap multiple MODIS pixels within a surrounding 3x3 km grid. As a result, $W_C$ is then calculated for each CloudSat pixel by averaging the nearest nine non-zero MOD-06-1KM (Platnick et al. 2003) pixels within a 3x3 grid surrounding each CloudSat pixel, which have been previously matched to the CloudSat track in the MOD-06-1KM product (Cronk and Platnick, 2018). There could be concerns that the averaging $W_C$ within the nearest nine MODIS pixels may not properly represent the $W_C$ at the appropriate scales relative to the horizontal footprint of each CloudSat pixel, however we tested our results using $W_C$ within the nearest MODIS pixel and found that our overall results do not change."*.

2. The authors state that "prior studies [of biases in MODIS cloud water] have found them to be small in comparison to other satellite retrievals". I suspect that this result may be resolution dependent and that in fact uncertainties for cloud smaller than several km in scale may be quite significant. For example, Cho et al. (2015) find that the MODIS cloud property retrievals from which the cloud water path is derived can have substantial errors in cumulus cloud fields because of partially cloudy pixels and horizontal homogeneity of cloud properties within the satellite footprint. Can the authors provide some greater support for the notion that the cloud water values are representative of the true value at the scales on the small end of the spectrum shown in this analysis?

a. **Thank-you for pointing out Cho et al. (2015) and that failure rates are higher in regions of broken cumulus. This should have been highlighted and caveated in the manuscript. Therefore we modified the following text (Pages 4-5, Lines 121-131) to account for this:**
*"Cho et al. (2015) found that MODIS effective radius and optical depth retrieval failure rates are higher in regions of broken trade cumulus than regions of predominantly stratocumulus, and they primarily attributed this to the presence of partially filled and inhomogeneous cloudy pixels. They also found that a large fraction of unexplained MODIS retrieval failures are related to the presence of precipitation after comparing MODIS failure rates to non-precipitating and precipitating pixels classified by CloudSat. This is attributed to a higher frequency of failures due to effective radius being too large. Considering the retrieval of effective radius and optical depth are required to derive $W_C$ and higher failure rates within broken trade cumulus, we suspect unavoidable sampling bias exists in $W_C$ matched to the smallest cloud objects and/or those containing large droplets and heavy rain. However on a global scale, prior studies have found the uncertainties in MODIS $W_C$ are small in comparison to other satellite retrievals (Seethala and Horvath, 2010; Lebsock and Su, 2014), with the global mean of MODIS $W_C$ being within 5 g m$^{-2}$ of $W_C$ determined using the Advanced Microwave Scanning Radiometer for Earth Observing System (AMSR‑E) (Seethala and Horvath, 2010)."*.

3. Fine resolution satellite imagery indicates that warm cumulus clouds substantially smaller than 1.7 km are common and in fact may be more prevalent than clouds larger than 1.7 km (e.g. Mieslinger et al. 2019). Presumably some of these clouds may be precipitated. Obviously, comparable data to the CloudSat data are not available at smaller scales from satellites. Nevertheless, do the authors expect that there may be a substantial population of precipitating cumulus clouds that are not captured in their analysis? Furthermore, one might expect that warm cumulus clouds might be limited inscale. Assuming crudely that cumulus clouds typically have an aspect ratio of around 1, one might presume that cumulus clouds broader than 5-10 km might also be tall enough to contain ice or mixed phase microphysical processes occurring. What characteristics ensure that the clouds included here are both warm liquid phase and truly cumulus clouds, or is the analysis expecting to include some stratocumulus clouds as well?

a. **The reviewer is correct in assuming that there is likely a large population of raining shallow cumulus smaller than CloudSat can detect leading to non-uniform beam filling. To address this (Pages 9-10, Lines 288-294) see the following text:** *"At the small end of the shallow cumulus horizontal size spectrum, CloudSat is limited to observing cloud objects no smaller than 1.4 x 1.8 km. Given prior ground observational studies, it is likely that there is a significant population of shallow cumulus cloud objects not identified by our study (e.g. Kollias et al., 2003; Mieslinger et al., 2019) due to non-uniform beam filling effects. Battaglia et al. (2020) noted that this results in an underestimation of path integrated attenuation, potentially introducing error into the retrieval of $W_p$. Unfortunately, this limitation is unavoidable given CloudSat's horizontal resolution."*.

b. **To address the potential issue of mixed phase clouds, we now explain in the following text (Page 4, lines 104-107) how we ensure that our analysis only includes warm cloud objects** *"To ensure that none of the cloud objects examined here contain ice, we only include cloud objects with tops entirely below the freezing level as defined in 2C-PRECIP-COLUMN Haynes et al. 2009)."*

4. The authors use the ratio of precipitation water to cloud water as their measure of"warm rain efficiency". Although, as the authors note, this quantity is just a proxy for the true efficiency. I think the authors are correct to make this point clear. I also think that perhaps it would be helpful for the authors to clarify what defines a proper quantitative measure of the warm rain efficiency. Presumably, it is not so easily observed, which is why they have chosen a proxy, which is fine. Given the brevity of this paper, however,I think a short elaboration on this point would be helpful. Furthermore, if the ratio used in this paper is merely a proxy for the true efficiency, is it really appropriate to be using "warm rain efficiency" throughout the manuscript to refer to this quantity? I suggest that the authors perhaps consider a different name so that readers are not confused about what is the true measure of the efficiency and what is the approximation of it. Alternatively, if there is a quantitative comparison of the ratio to the true efficiency, perhaps from a theoretical study, then it might be appropriate to refer to the proxy value as a measure of the efficiency with some quoted uncertainty value.

   a. **We agree that we should have defined warm rain efficiency in a proper context, therefore we added the following text (Pages 2-3, Lines 51-59) to the paper:** *"Prior studies have defined precipitation efficiency in two ways: 1) as the large-scale precipitation efficiency and 2) as the cloud microphysical precipitation efficiency. Generally, observational studies have based their definition of precipitation efficiency on the large-scale definition, which has simply been defined as the ratio of surface rain rate to the sum of both vapor mass flux in/out of a cloud and surface evaporation (e.g. Chong and Hauser, 1989; Tao et al., 2004; Sui et al., 2007), whereas the cloud microphysical definition, or the ratio of surface rain rate to the sum of vapor condensation and deposition rates, has been primarily used in cloud modeling studies (e.g. LI et al., 2002; Sui et al., 2005; Gao et al., 2018). Although both the large-scale and cloud microphysical definitions of precipitation efficiency are useful (Sui et al., 2005; Sui et al., 2007), variations in the ratio of cloud water to rain water (WRR) in response to changes in evaporation can theoretically be used as a proxy for warm rain efficiency based on the cloud microphysical definition."* **Additionally, we changed any reference to WRE, in the context of this paper, to the ratio of cloud water to rain water (WRR) as well as** *"warm rain efficiency"* **in the title to** *"the ratio of cloud water to rain water".*

5. The corroboration of the inferences based on the ratio of precipitating water to cloudwater with the inferences from the vertical gradient in reflectivity (VGZ) is a valuable contribution of this paper and certainly strengthens the case that the authors are making. In lines 174 to 180 the authors argue that the dependence of VGZ on cloud-top height supports the notion that updrafts in larger clouds are protected from entrainment. Why would this dependence on cloud-top height not simply result from collision/coalescence happening through a deeper cloud layer independent of any difference in entrainment? Presumably the taller clouds are provide a broader distance from cloud base to cloud top through which raining drops can fall and collect cloud drops. Likewise, perhaps a stronger updraft that yields a taller cloud is better at promoting the coalescence of cloud drops through turbulent collisions. Could these similarly explain the differences between clouds of differing heights?

a. Yes, these factors could also explain differences in VGZ as a function of extent as well as differences in the ratio of cloud water to rain water for cloud objects with different heights. To address this we added the following text in a section called *"Limitations of analysis and observations"* to our paper (Page 9, Lines 268-287) *"This study has emphasized the potential for the decreasing impact of entrainment on cloud cores, resulting in higher WRR, as cloud size increases; however, it is important to point out other factors related to cloud size that may also impact WRR. Figure 3 shows WRR is higher when cloud objects are taller, which may be simply because we are sampling more mature clouds that have had more time for the collision-coalescence process to result in rain formation. Deeper shallow cumulus not only live longer which would give cloud droplets more time to grow to raindrop size (e.g. Burnet and Brenguier, 2010), but they are more likely to have more intense updrafts which could result in more water vapor being transported to higher altitudes within a cloud. Stronger updrafts are then more likely to be able to suspend cloud droplets higher in the cloud for longer periods of time which allows them to grow larger before they begin to fall and collision-coalescence is initiated. Once cloud droplets do begin to fall, they are not only potentially larger but able to collect more droplets over a larger distance than droplets falling through a shallower cloud. This could potentially result in higher WRR, however there is likely a lag between the peaks in cloud water path and rain water path as cloud drops grow to raindrop size in a developing cloud. Earlier modeling studies have also noted that turbulent flow potentially enhances the likelihood of warm rain formation (e.g. Brenguier and Chaumat, 2001; Seifert et al., 2010; Wyszogrodzki et al., 2013; Franklin, 2014; Seifert and Onishi, 2016; Chen et al., 2018). Seifert et al. (2010) found that turbulence effects are largest near cloud tops in shallow cumulus, which they note is an important region for initial rain formation. While these additional processes may impact WRR, the satellite observations used in this study are instantaneous snapshots in time. We attempted to remove some of these life cycle impacts by binning cloud objects by top height. Within a given cloud top height bin, WRR (Figure 3) and the magnitude of $VGZ_{CP}$ (Figure 4c) still increase as a function of extent. While we acknowledge that this cannot fully remove these impacts, these results support the idea that processes other than those related to cloud lifetime, like lateral entrainment, may also influence*

*the WRR of shallow cumulus of different horizontal sizes".*

6. Finally, the authors explore the dependence of their proxy for warm rain efficiency on the aerosol optical thickness in the vicinity of the cloud. They conclude that there is little dependence of the efficiency on aerosols, which is an interesting result. I suggest, though, that the authors remove the word "surprisingly" from the abstract where this result is reported. As noted by the authors, by excluding non-precipitating clouds from their analysis they are likely missing the expected dominant effect, which is the suppression of rain formation. Is there not a CloudSat study looking at the dependence of the occurrence of rain in CloudSat retrievals upon AOD? I think that a citation to such a study would be appropriate in the discussion of the results presented in this paper. If not, I think the authors should point out that this might be the more fruitful path to quantifying aerosol effects.

   a. **We have removed "surprisingly" from the abstract**
   b. **To address the second part of your comment regarding the dependence of the occurrence of rain in CloudSat retrievals upon AOD, we added a figure (Figure 5d) which shows the rain likelihood determined using CloudSat cloud objects at a given AOD. For reference, it is described in the following text on Pages 8-9, Lines 261-264:** *"Figure 5d shows the likelihood of rain occurrence at a given AOD determined by the ratio of raining cloud objects to the total number of cloud objects. As expected, Figure 5d shows that the likelihood of rain decreases as AOD increases, with rain likelihood of about 50% in the cleanest environments decreasing to about 40% for an AOD approaching 0.75. These results imply that once the condensation-coalescence is initiated, aerosol loading has a smaller impact on the conversion of cloud water to rain than other cloud or environmental characteristics."*.

**References:**

Cho, H.-M., Zhang, Z., Meyer, K., Lebsock, M., Platnick, S., Ackerman, A. S., Di Girolamo, L., C.-Labonnote, L., Cornet, C., Riedi, J., and Holz, R. E.: Frequency and causes of failed MODIS cloud property retrievals for liquid phase clouds over global oceans, Journal of Geophysical Research: Atmospheres, 120, 4132–4154, https://doi.org/https://doi.org/10.1002/2015JD023161, https://agupubs.onlinelibrary. 390 wiley.com/doi/abs/10.1002/2015JD023161, 2015.

---

## Author Response (AR2)

**Reviewer 1**

We thank Reviewer 1 for all the help in improving this manuscript and your minor comments are addressed below in **red**:

The authors have made significant efforts to address my comments, which I appreciate. The thorough replies and tracked changes were also much appreciated. I think the paper is improved and ready to be published.

I have just a couple minor comments to consider. There are some missing citations (e.g., on lines 62-63 and 111) that need to be fixed. In addition, with the added text, some paragraphs are now very long as well, which could potentially be broken up.

**We split a couple of the now long paragraphs in the manuscript (lines 34 – 63 and lines 95 – 116). However, we left both the methods paragraph focusing on cloud water content (117 – 145) and the first paragraph of section 4 (lines 269 – 288) the same, because there wasn't an obvious break-point in either paragraph without rewriting the text.**

**Both authors checked lines 62-63 and line 111 for missing references and could not find any. So, we are unsure if the reviewer's line numbers are off, or if they think there are specific studies not cited that should be cited. Could you clarify what references are missing?**

**Reviewer 2**

We thank the reviewer for all their help in improving this manuscript.

The two principal issues raised in the review of this paper are: 1) the limitations of the satellite data; mainly related to uncertainties in sampling small clouds, and 2) the interpretation of the results as driven by impacts of entrainment on precipitation formation to the possible exclusion of other potential processes. I feel that the authors have done a suitable job of responding to these concerns from the reviewers, acknowledging them in their revised manuscript, and defending their arguments in light of these uncertainties. Thus, I feel that this paper is suitable for publication.

[revised manuscript text omitted]